# Mitochondrial metabolism sustains CD8+ T cell migration for an efficient infiltration into solid tumors

Luca Simula [1] ✉, Mattia Fumagalli[1], Lene Vimeux[1], Irena Rajnpreht[1], Philippe Icard[2,3], Gary Birsen [4], Dongjie An[1], Frédéric Pendino[1], Adrien Rouault[1], Nadège Bercovici [1], Diane Damotte [5], Audrey Lupo-Mansuet [5], Marco Alifano[3,6], Marie-Clotilde Alves-Guerra [7] & Emmanuel Donnadieu [1]✉

The ability of CD8+ T cells to infiltrate solid tumors and reach cancer cells is associated with improved patient survival and responses to immunotherapy. Thus, identifying the factors controlling T cell migration in tumors is critical, so that strategies to intervene on these targets can be developed. Although interstitial motility is a highly energy-demanding process, the metabolic requirements of CD8+ T cells migrating in a 3D environment remain unclear. Here, we demonstrate that the tricarboxylic acid (TCA) cycle is the main metabolic pathway sustaining human CD8+ T cell motility in 3D collagen gels and tumor slices while glycolysis plays a more minor role. Using pharmacological and genetic approaches, we report that CD8+ T cell migration depends on the mitochondrial oxidation of glucose and glutamine, but not fatty acids, and both ATP and ROS produced by mitochondria are required for T cells to migrate. Pharmacological interventions to increase mitochondrial activity improve CD8+ T cell intratumoral migration and CAR T cell recruitment into tumor islets leading to better control of tumor growth in human xenograft models. Our study highlights the rationale of targeting mitochondrial metabolism to enhance the migration and antitumor efficacy of CAR T cells in treating solid tumors.

Although T cell-based immunotherapies have transformed cancer treatment, only a minority of patients with solid cancers responds to these approaches[1]. Several mechanisms have been proposed to explain the reduced CD8+ T cell function within the tumor microenvironment (TME)[2,3], including a defective CD8+ T cell intratumoral motility. Indeed, for an effective direct destruction of cancer cells, CD8+ T cells must fulfill an active migration allowing them to make contact with malignant cells[4]. In most human solid tumors, tumor cells lie within tumor islets surrounded by a stroma composed of fibers and host-derived suppressive cells[5–7]. Remarkably, accumulating evidence suggests that such TME structure limits CD8+ T cells from migrating and contacting their targets. We previously reported that several TME elements, such as pro-tumor macrophages and extracellular matrix fibers, can sequester CD8+ T cells in stromal areas, thus preventing

[1]Institut Cochin, Inserm U1016, CNRS UMR8104, Université Paris-Cité, Equipe labellisée "Ligue contre le Cancer", Paris 75014, France. [2]Université de Normandie, UNICAEN, Inserm U1086 Interdisciplinary Research Unit for Cancer Prevention and Treatment, Caen, France. [3]Thoracic Surgery Department, Cochin Hospital, APHP-Centre, Université Paris-Cité, Paris, France. [4]Department of Pneumology, Thoracic Oncology Unit, Cochin Hospital, APHP-Centre, Université Paris-Cité, 75014 Paris, France. [5]Department of Pathology, Cochin Hospital, APHP-Centre, Université Paris-Cité, 75014 Paris, France. [6]Inserm U1138, Integrative Cancer Immunology Unit, 75006 Paris, France. [7]Institut Cochin, Inserm U1016, CNRS UMR8104, Université Paris-Cité, Paris 75014, France. ✉e-mail: luca.simula@inserm.fr; emmanuel.donnadieu@inserm.fr

their contact with tumor cells[8–10]. Interestingly, such a hostile TME can also prevent the infiltration of CD8[+] CAR T cells[11,12], thus limiting the effectiveness of immunotherapy approaches based on CAR T cell infusion for solid cancer patients. Beyond extracellular determinants, intracellular factors also play an important role in the ability of T cells to migrate and reach tumor cells.

Metabolism sustains T cell migration, which relies on a fine-tuned regulation of cytoskeleton remodeling for which metabolic reactions supply energy in the form of ATP and GTP (for actin and tubulin polymerization). Although ATP is known to support both fast-directional migration[13] and slow motility[14] of activated T cells and inhibition of its production reduces T cell motility[15], very few studies have so far addressed how specific metabolic pathways support T cell migration. Some of them highlighted the role of OXPHOS-produced ATP[13,16], others that of glycolysis[15,17,18], whose end-product (lactate) inhibits T cell motility[15]. However, the relative importance of these metabolic pathways in sustaining T cell motility has never been addressed. In addition, most of these studies have been carried out using 2D models (mainly transwell assay or protein-coated surfaces)[13–16], which are not suited to recapitulate the 3D amoeboid-like motility of CD8[+] T cells within the TME. This aspect is important since due to nutrients unavailability, the accumulation of waste products, the acid pH and the low oxygen tension, CD8[+] T cell metabolism can be severely altered within TME[1,19]. Such alterations have already been associated with a defective cytotoxicity, persistence and survival of CD8[+] T cells, favouring tumor escape[1]. Since motility is a highly energy-demanding process, it is extremely likely that these metabolic alterations could also impair CD8[+] T cell intratumoral motility, thus playing an important role in preventing these cells from reaching and killing tumor cells. However, our lack of knowledge about how metabolic pathways support CD8[+] T cell 3D motility within the TME prevents us from anticipating how TME metabolic alterations could affect T cell motility and developing effective strategies to overcome these defects, especially for CAR T cell-based therapy.

Here, we investigate how metabolism supports CD8[+] T cell 3D motility using relevant 3D motility models. Our data indicate that (i) an effective CD8[+] T cell 3D motility is supported mainly by glucose- and glutamine-fueled TCA cycle sustaining both ATP and mtROS production from mitochondria and (ii) strategies targeting mitochondrial metabolism are effective in increasing the intratumoral infiltration of CD8[+] CAR T cells to help them reach and kill tumor cells in preclinical solid tumor models.

## Results

### Human CD8[+] T cell 3D motility is mainly supported by the TCA cycle fueled by glucose and glutamine

To investigate the metabolic regulation of CD8[+] T cell motility in a 3D environment, we evaluated the spontaneous migration of activated CD8[+] T cells in a 3D collagen gel using time-lapse imaging microscopy. Naïve CD8[+] T cells were activated (anti-CD3/CD28) and expanded in the presence of IL7 + IL15 cytokines (Fig. 1A). First, we tested the role of glucose and glutamine, the main energy sources in culture media. During acute deprivation (1 h), the absence of glucose, but not glutamine, slightly but significantly reduced 3D motility (Fig. 1B). This effect was presumably due to the different concentrations of these nutrients in the culture medium, since increasing glutamine concentration to match glucose one abrogated this difference (Fig. 1C). After prolonged deprivation (48 h), both nutrients halved 3D motility in a similar way (Fig. 1D). These data suggest that glucose and glutamine support similarly 3D motility as they can compensate for each other during a short time, while for longer periods both are similarly required, like it was observed for T cell proliferation (Supplementary Fig. 1A). Interestingly, while only glucose sustains glycolysis, both nutrients may support similarly TCA cycle in T cells, as suggested by $CO_2$ flux analysis (Supplementary Fig. 1B). We excluded global changes in the T cell

population in terms of mitochondrial mass, mitochondrial membrane potential (MMP), mtROS amount and GLUT1 expression following glucose or glutamine deprivation (Supplementary Fig. 1C). Next, we investigated whether fatty acids (FAs) also sustain 3D motility. Since FAs are almost not present in culture media, we added oleic acid, linoleic acid, or palmitate during the motility assay. Surprisingly, although these FAs have been reported to modulate some functions in T cells[20–24], we observed no effect on motility (Fig. 1E). In line with this, FAO inhibitors etomoxir, perhexiline (targeting Carnitine Palmitoyl-transferase IA, CPT1A) or trimetazidine (targeting 3-Ketoacyl-CoA Thiolase enzyme, 3-KAT) had no impact on motility (Fig. 1F). Overall, these data indicate that glucose and glutamine, but not FAs, are required to support CD8[+] T cell 3D motility under normal conditions.

We next decided to evaluate the impact of pharmacological inhibitors of glycolysis (2DG), TCA cycle (6,8-Bis(benzylthio)-octanoic acid [6,8bOA], an inhibitor of pyruvate dehydrogenase [PDH] and oxoglutarate dehydrogenase [OGDH]) and glutaminolysis (CB-839 for glutaminase-1 [GLS1] and Compound-968 for GLS2) (see scheme in Supplementary Fig. 1D). First, we observed that inhibition of GLS1-mediated glutaminolysis significantly reduced 3D motility (Fig. 1G). Although the GLS2 enzyme is also present in activated CD8[+] T cells, it is only expressed at very low levels (see Schmiedel Dataset at The Human Protein Atlas). Consistent with this, we found that, in contrast to what we observed in presence of GLS1 inhibitor CB-839, the GLS2 inhibitor Compound-968 had no impact on mitochondrial respiration (Supplementary Fig. 1E) or 3D motility in a collagen gel (Supplementary Fig. 1F), suggesting that GLS2 is dispensable for these functions (this might be due to its low expression level and/or the ability of GLS1 to compensate for its activity). The inhibition in cell motility due to GLS1 inhibitor CB-839 was even more pronounced in a medium containing no glucose (Supplementary Fig. 1G). Second, 3D motility was similarly reduced in the presence of 2DG (which inhibits both glycolysis and glucose-derived OXPHOS) or etomoxir + CB-839 combination (whose effect is mainly due to inhibition of glutaminolysis, since etomoxir alone had no effect) (Fig. 1H and Supplementary Movies 1–6). Third, a much stronger decrease in 3D motility was observed in the presence of TCA cycle inhibitor 6,8bOA (Fig. 1H and Supplementary Movies 1–6). Fourth, a complete blockade of the TCA cycle by the triple combination of 6,8bOA + etomoxir + CB-839 (allowing only glycolysis to generate energy) completely blocked 3D motility (less than 10% of initial motility left) (Fig. 1H and Supplementary Movies 1–6). We excluded that these effects were due to global changes in the T cell population in terms of mitochondrial mass, MMP, mtROS amount and GLUT1 expression (Supplementary Fig. 1H). The only exception was an increase in mtROS amount upon 6,8bOA treatment, which has previously been described as one of the mechanisms by which this drug inhibits OGDH enzyme[25]. Taken together, these data suggest that CD8[+] T cell 3D motility is mainly supported by the TCA cycle while glycolysis plays a minor role. This is in clear contrast to what is known for T cell proliferation, which relies more on glycolysis-derived energy[26,27], as we also observed (Supplementary Fig. 1I). Of note, these findings are not limited to cells expanded in the presence of IL-7 + IL-15 (which promotes mitochondrial metabolism[28]), since similar results were also observed in cells cultured with IL-2 (more glycolysis-prone[29]) (Supplementary Fig. 1J, K). Next, to validate the effects of 6,8bOA in a more relevant TME-like model, we injected fluorescence-labeled activated CD8[+] T cells into a fresh tumor fragment obtained from s.c. inoculation of human BxPC3 pancreatic cells into NSG nude mice (termed BxPC3-NSG model). Then, the tumor piece was cut into viable slices and the motility of CD8[+] T cells was recorded with a spinning disk confocal microscope. Interestingly, slices incubated with 6,8bOA significantly reduced CD8[+] T cell 3D motility (Fig. 1I and Supplementary Movies 7, 8). To confirm these effects, we took advantage of CRISPR/Cas9 approach to edit the 6,8bOA targets, i.e., PDH and OGDH enzymes. Surprisingly, although efficiently edited (Supplementary Fig. 1L),

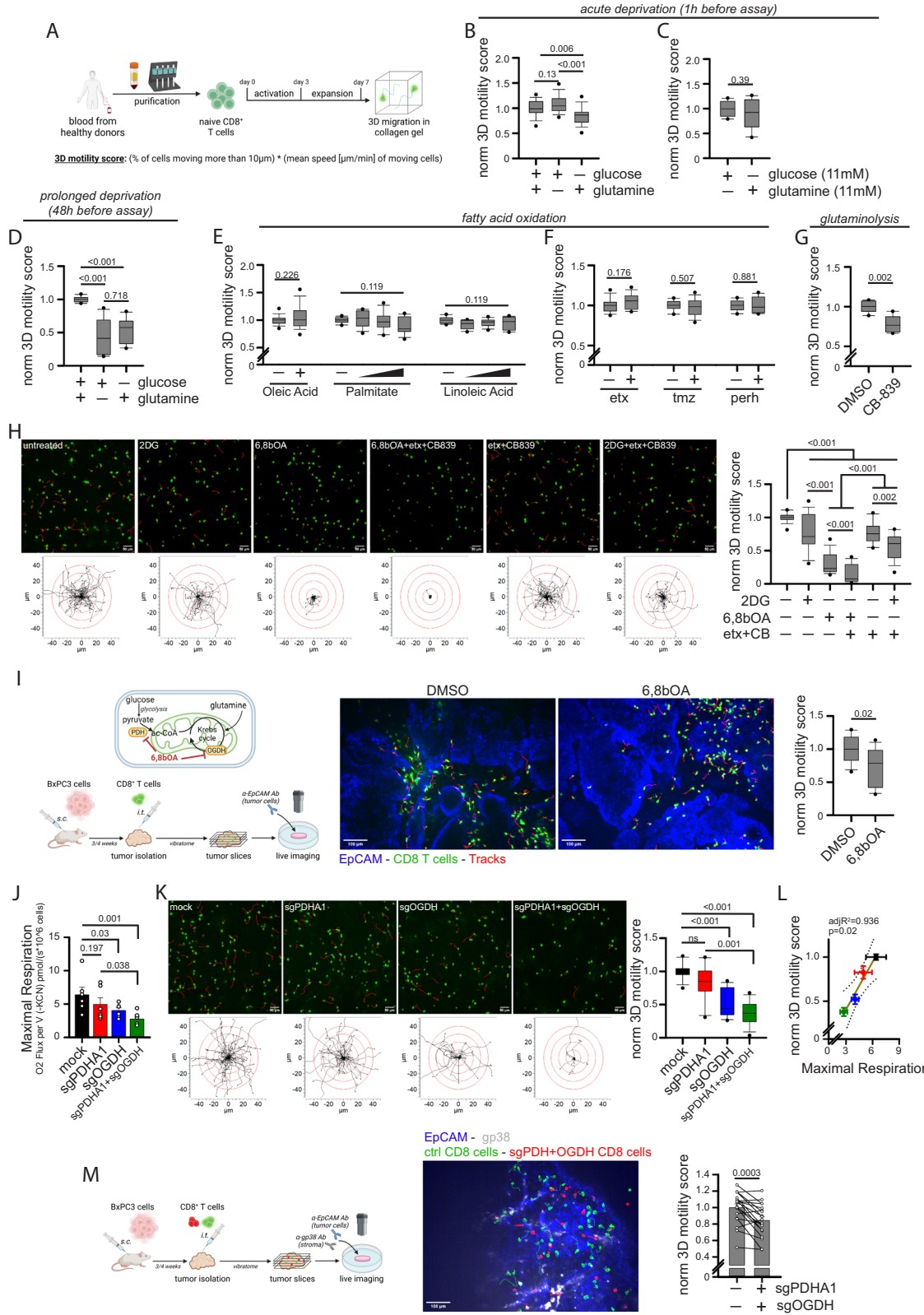

down-regulation of the *PDHA1* gene in T cells (*PDHA2* is not expressed) had no impact on mitochondrial respiration (Fig. 1J and Supplementary Fig. 1M) nor 3D motility (Fig. 1K and Supplementary Movies 9–12), indicating the presence of compensatory mechanisms to maintain TCA cycle activity. In contrast, *OGDH* down-regulation (alone or in combination with *PDHA1* editing) (Supplementary Fig. 1L) significantly reduces both maximal respiration (Fig. 1J) and 3D motility in CD8+

T cells (Fig. 1K and Supplementary Movies 9–12), an effect also confirmed using the OGDH inhibitor succinyl-phosphonate (Supplementary Fig. 1N). Remarkably, under these conditions, we observed an almost perfect correlation between maximal mitochondrial respiration and 3D motility in collagen gel (Fig. 1L), further corroborating the notion that mitochondrial metabolism is the main player supporting 3D motility. Furthermore, CRISPR/Cas9-mediated double *PDHA1/OGDH*

**Fig. 1 | CD8⁺ T cell 3D motility is mainly supported by the TCA cycle fueled by glucose and glutamine but not fatty acids. A** Experimental layout. **B–H** Normalized 3D motility Score of activated CD8⁺ T cells in a motility medium (see Methods) containing no glucose or no glutamine (**B**, $n = 27$ movies, One-Way ANOVA with posthoc Holm Sidak's), either glucose or glutamine at 11 mM (**C**, $n = 10$ movies, unpaired two-tailed Student's T-Test), starved from glucose or glutamine since 48 h before evaluating motility (**D**, $n = 12$ movies, ANOVA on Ranks with posthoc Tukey's; also the motility assay was performed in the absence of glucose or glutamine) or in presence of the indicated nutrients or drugs (2DG: 2-deoxy-glucose; etx: etomoxir; tmz: trimetazidine; perh: perhexiline; 6,8bOA: 6,8-bis(benzylthio)octanoic acid; CB: CB-839) (**E**, oleic acid $n = 29$ movies [unpaired Student's T-Test], linoleic acid & palmitate $n = 12$ movies [ANOVA on Ranks]; **F**, etx $n = 12$ movies, tmz & perh $n = 10$ movies [unpaired two-tailed Student's T-Test]; **G**, $n = 6$ movies [unpaired two-tailed Student's T-Test]; **H**, $n = 18$ movies [ANOVA on Ranks with posthoc Student–Newman–Keuls's]). In (**H**), (top) 2D z-stack reconstructions of 3D motility in collagen gel (cells in green, tracks in red) and (bottom) superimposed tracks normalized to their starting coordinates (tracks in black, red circles every 10 μm) are shown for each condition. **I** Experimental layout (bottom) and enzymes inhibited by 6,8bOA (top) are shown on the left. 3D motility of Calcein

Green-stained activated CD8⁺ T cells (cells in green, tracks in red) within viable tumor slice (tumor cells in blue, EpCAM) derived from the BxPC3/NSG model and incubated with the indicated drugs ($n = 12$ slices, unpaired two-tailed Student's T-Test). **J, L** Measurement of maximal respiration (**J**, $n = 6$ biologically independent samples, One-Way ANOVA on Repeated Measurements with posthoc Holm Sidak's) and normalized 3D motility Score (**K**, $n = 15$ movies, ANOVA on Ranks with posthoc Tukey's) and correlation between the two parameters (**L**, Linear Regression Test from mean values of (**J, K**), linear regression mean ± 95% confidence interval are reported in yellow and gray, respectively) in activated CD8⁺ T cells CRISPR/Cas9-edited for the indicated genes. **M** Experimental layout on the left. 3D motility of activated Calcein Green-stained control CD8⁺ T cells (green) and Calcein Red-stained CRISPR/Cas9-edited CD8⁺ T cells (sgPDH+sgOGDH) (red) within viable tumor slice (tumor cells in blue, EpCAM; stroma in gray, gp38) derived from the BxPC3/NSG model ($n = 19$ slices, paired Student's T-Test). Data are expressed as mean ± SEM in (**J, L, M**), and as box plot (center line, median; box limits, upper and lower quartiles; whiskers, 1.5x interquartile range; points, 5th and 95th percentiles) in (**B–I, K**). Scale bar, 50 μm in (**H, K**) and 100 μm in (**I, M**). Schemes in (**A, I, M**) have been created with BioRender.com.

knock-down significantly reduced CD8⁺ T cell intra-tumoral motility as assessed using tumor slice derived from the BXPC3-NSG model (Fig. 1M and Supplementary Movie 13). Last, we observed that CD8⁺ T cell 3D motility in collagen gel can be increased by forcing glucose-derived pyruvate to enter TCA cycle through administration of (*i*) lactate dehydrogenase and PDH kinase pharmacological inhibitors (LDHi and PS10, respectively) or (*ii*) a PDH activator (α-lipoic acid) (Supplementary Fig. 1O, P).

Overall, these data support the hypothesis that CD8⁺ T cell 3D motility is mainly supported by mitochondrial metabolism, and more specifically by the TCA cycle fueled by glucose and glutamine.

## Glucose fuels human CD8⁺ T cell 3D motility mainly via the TCA cycle and to a lesser extent via glycolysis

Our data (Fig. 1) suggested that glucose sustains CD8⁺ T cell motility mainly via OXPHOS and poorly via glycolysis. To further clarify the relative importance of these two pathways in supporting CD8⁺ T cells 3D motility, we performed the following experiments.

First, to force cells to rely on OXPHOS-derived ATP, we cultured CD8⁺ T cells in the presence of glucose or galactose for 3 days. Of note, galactose cannot provide ATP via glycolysis but only via OXPHOS[30,31]. While 3D motility was minimally impacted by the replacement of glucose by galactose (Fig. 2A and Supplementary Movies 14, 15), proliferation was strongly impaired (Fig. 2B), consistent with the notion that T cell proliferation, but not motility, requires glycolytic ATP.

Second, we cultured CD8⁺ T cells in the presence of a low dose of ethidium bromide (EtBr) to deplete mtDNA[32], forcing the cells to rely on glycolytic ATP. Two-week EtBr treatment significantly reduced the amount of mtDNA (Fig. 2C) and the mitochondrial respiration while increasing glycolysis (Fig. 2D, see also Supplementary Fig. 2A, B) and mtROS levels (Supplementary Fig. 2C). In contrast to what was observed with galactose, EtBr-treated CD8⁺ T cells showed a strong reduction in 3D motility (Fig. 2E and Supplementary Movies 16, 17) but no effect on cell proliferation (Fig. 2F).

Third, we evaluated the ability of several nutrients to rescue motility when glycolysis or OXPHOS were inhibited in a medium containing only glucose and no glutamine. Under this condition, the addition of 2DG strongly inhibited 3D motility (Fig. 2G and Supplementary Movies 18–20). However, 3D motility was restored by the addition of lactate, which bypasses upstream glucose metabolism and directly fuels the mitochondrial TCA cycle (Fig. 2G and Supplementary Movies 18–20). To confirm that lactate is actively taken up by T cells to maintain motility upon glycolysis inhibition, we tested cell motility in presence of monocarboxylate transporter 1 (MCT1) inhibitor AZD3965

(MCT1i) and we found that this drug was indeed able to prevent lactate-mediated rescue of cell motility in the presence of 2DG (Supplementary Fig. 2D). In contrast, lactate failed to rescue the inhibition of 3D motility due to TCA-cycle inhibitor 6,8bOA (Fig. 2H and Supplementary Movies 21–23) or OXPHOS inhibitor oligomycin (Fig. 2I and Supplementary Movies 24–26). A similar rescue of 2DG- but not oligomycin-mediated inhibition was observed using additional nutrients downstream of glycolysis (pyruvate and acetate) and cell-permeable derivatives of TCA cycle intermediates (dimethyl-succinate and dimethyl-α-ketoglutarate) (Supplementary Fig. 2E, F). Of note, the ability of lactate to rescue 2DG- but not oligomycin-induced reduction in motility was also observed in tumor slices derived from the BxPC3-NSG model (Fig. 2J and Supplementary Movies 27–32). Last, the administration of nutrients such as lactate, acetate, or pyruvate increased 3D motility even in complete medium containing both glucose and glutamine (Supplementary Fig. 2G), presumably by providing an extra boost to TCA cycle and OXPHOS machinery.

In sum, these data suggest that (*i*) glucose sustains CD8⁺ T cell 3D intratumoral motility mainly via OXPHOS and not glycolysis and (*ii*) several nutrients (including lactate) could be used by CD8⁺ T cells to support their motility in the absence of glucose.

## Human CD8⁺ T cell mitochondrial metabolism and 3D motility are correlated in different contexts

Given the important role of mitochondrial metabolism in supporting CD8⁺ T cell 3D motility, we wondered whether these two parameters correlate in CD8⁺ T cells under different conditions.

First, we decided to compare CD8⁺ T cells expanded in the presence of IL-2 or IL-7 + IL-15 (IL-7/15), since these cytokines differentially affect the glycolysis/OXPHOS rate in T cells[28,29]. IL-7/15 CD8⁺ T cells showed a similar glycolytic rate but increased mitochondrial respiration (Fig. 3A and Supplementary Fig. 3A, B) and higher mitochondrial mass and glucose uptake with similar levels of mitochondrial membrane potential (MMP) (Supplementary Fig. 3C) compared to IL-2 cells. In line with our data, such increased mitochondrial activity correlated with a superior 3D motility in collagen gel (Fig. 3B and Supplementary Movies 33, 34). Under these conditions, we also found that 3D motility correlated strongly with several OXPHOS parameters, such as mitochondrial mass, mitochondrial membrane potential (MMP) and expression of the TCA-cycle enzyme isocitrate-dehydrogenase-2 (IDH2), as well as with global glucose uptake and expression of glucose receptor GLUT1 (Supplementary Fig. 3D). On the contrary, 3D motility was not or negatively correlated with the expression level of glycolytic enzymes phospho-fructokinase-2 (PFK2) and hexokinase-1 (HK-1), as well as of the monocarboxylate transporter-1 (MCT-1) and

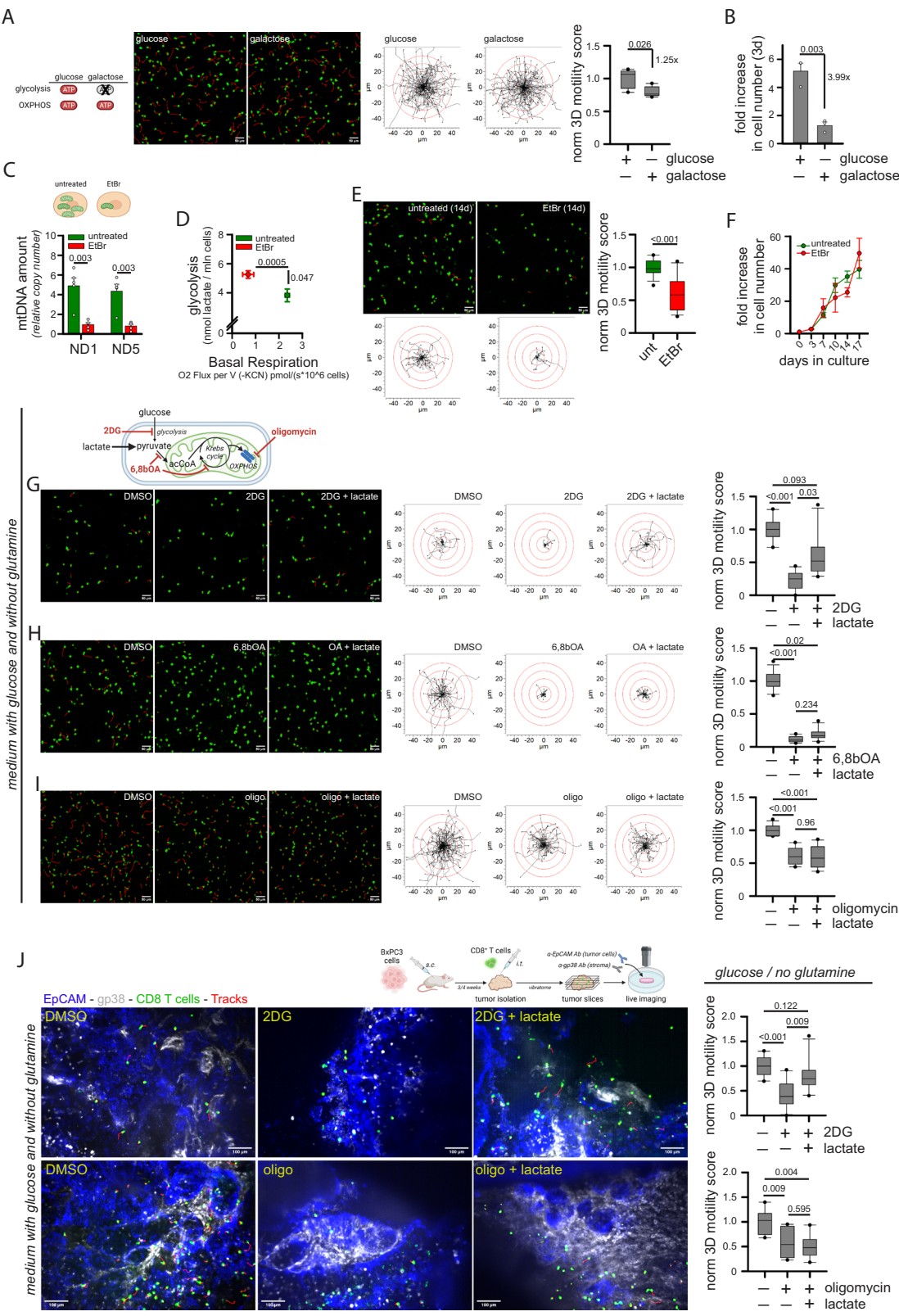

the pentose-phosphate pathway (PPP) enzyme glucose-6-phosphate-dehydrogenase (G6PD) (Supplementary Fig. 3D). The higher motility of IL7/15 CD8[+] T cells compared to IL2 cells was also confirmed in tumor slices derived from the BxPC3-NSG model (Fig. 3C and Supplementary Movie 35). This enhancement was mainly due to IL-15 (Supplementary Fig. 3E, F), in line with its prominent role in supporting mitochondrial metabolism[28].

Second, to compare T cell motility in the absence of culturing cytokines, we evaluated 3D motility and the level of several metabolic parameters in freshly isolated human peripheral blood T (hPBT) cells stained with anti-CD8 and anti-CD45RA Abs to distinguish 4 populations (CD8 or CD4 either naïve-like 45RA[pos] or memory-like 45RA[neg]). Once again, we observed that the subpopulation migrating more efficiently (i.e., CD8[neg]45RA[neg] cells, CD4 memory-like) exhibited higher

**Fig. 2 | Glucose fuels CD8⁺ T cell 3D motility mainly through the TCA cycle and poorly through glycolysis. A, B** Normalized 3D motility Score (**A**, $n = 5$ movies, unpaired two-tailed Student's T-Test) and fold increase in cell number (**B**, $n = 3$ independent biological samples, unpaired two-tailed Student's T-Test) of activated CD8⁺ T cells cultured since 72 h before motility assay in medium containing 11 mM glucose or galactose. **C–F** Activated CD8⁺ T cells were cultured for at least 14d in control medium (untreated) or in the presence of a low dose of ethidium bromide (EtBr). After 14d, the following parameters were measured: mtDNA amount (**C**, $2^{-\Delta Ct}$ values between mtDNA genes ND1 or ND5 and nuclear gene 18 S; $n = 5$ independent biological samples, paired Student's T-Test), energy plot (**D**, raw measurements are reported in Fig. S2A, B, Linear Regression Test from mean values of S2A [$n = 3$ independent biological samples] and S2B [$n = 4$ independent biological samples]), normalized 3D motility Score (**E**, $n = 15$ movies, unpaired two-tailed Student's T-Test), fold increase in cell number (**F**, $n = 3$ independent biological samples, Two-Way ANOVA). **G–I** Normalized 3D motility Score of activated CD8⁺ T cells in the presence of 2-deoxy-glucose (2DG; **G**, $n = 10$ movies, ANOVA on Ranks with posthoc Tukey's), 6,8-bis(benzylthio)octanoic acid (6,8bOA; $n = 12$ movies, ANOVA on Ranks with posthoc Tukey's) and oligomycin (oligo; **I**, $n = 12$ movies, ANOVA on Ranks with posthoc Tukey's) with or without lactate in a medium containing only glucose and no glutamine. **J** Experimental layout on the top. 3D motility of Calcein Green-stained activated CD8⁺ T cells (cells in green, tracks in red) within viable tumor slices (tumor cells in blue, EpCAM; stroma in gray, gp38) derived from the BxPC3/NSG model and incubated with the indicated drugs or nutrients (top $n = 10$ movies, ANOVA on Ranks with posthoc Holm Sidak's; bottom, $n = 8$ movies, ANOVA on Ranks with posthoc Holm Sidak's). Data are expressed as mean ± SEM in (**B, C, D, F**), and as box plot (center line, median; box limits, upper and lower quartiles; whiskers, 1.5x interquartile range; points, 5th and 95th percentiles) in (**A, E, G–J**). Scale bar, 50 μm in (**A, E, G–I**) and 100 μm in (**J**). Schemes in (**A, C, G, J**) have been created with BioRender.com.

mitochondrial mass and MMP, while no selective increase in glycolysis-associated parameters (Supplementary Fig. 3G).

Third, we previously reported that CD8⁺ T cells localizing in tumor islets of non-small cell cancer (NSCLC) biopsies move faster than CD8⁺ T cells in the surrounding stroma[10], an observation which we confirmed (Fig. 3D). We therefore wondered whether this higher motility correlates with an increase in mitochondria-associated parameters. We observed that CD8⁺ T cells located within tumor islets showed higher expression of several mitochondrial proteins (TOM20, IDH2 and ATP5a) compared to CD8⁺ T cells located in the surrounding stroma (Fig. 3E). In contrast, no differences were observed for key glycolytic enzymes, such as HK1 and PFK2 (Fig. 3E). These data indicate that 3D motility and mitochondrial parameters correlate when comparing CD8⁺ T cell subpopulations infiltrating NSCLC tissue.

Last, we asked whether the correlation between motility and mitochondrial metabolism holds true also when comparing different cells within the same CD8⁺ population, *i.e.*, if the cells moving faster also show higher mitochondrial activity. To answer this point, we evaluated 3D motility of CD8⁺ T cells stained with TMRE or MitoSox (to monitor MMP and mtROS respectively, two proxies of mitochondrial activity in cells). As shown in Fig. 3F, G, CD8⁺ T cells migrating faster also showed higher levels of MMP and mtROS amount compared to slower cells.

Overall, our observations suggest that mitochondrial metabolism and 3D motility are positively correlated in CD8⁺ T cells in multiple contexts.

## TCA cycle sustains ATP and ROS production from mitochondria to support human CD8⁺ T cell 3D motility

The highly energetic process of CD8⁺ T cell migration in 2D[13,14] and 3D environments helps explain the high requirement for mitochondria, which are the main sources of cellular ATP. Indeed, CD8⁺ T 3D motility is progressively more inhibited after 1 h of incubation with increasing doses of oligomycin (Fig. 4A). However, we noticed a strong reduction in 3D motility between 1 μM and 10 μM oligomycin doses, which cannot be explained by differences in ATP production, since OXPHOS was already efficiently blocked by 1 μM of oligomycin (Fig. 4B). Both doses showed no toxic effects on mitochondrial functioning, since mitochondrial respiration correctly re-increased after CCCP administration, indicating unaltered ETC functionality (Supplementary Fig. 4A, B). Interestingly, we noticed an abrupt decrease in mtROS amount between these two doses of oligomycin, correlating with the reduction in 3D motility (Fig. 4C) and suggesting that also mtROS might positively sustain 3D motility. ROS have pleiotropic roles in the regulation of cell migration[33–36], particularly the amoeboid-like motility typical of T cells within 3D environment[37,38]. To better clarify the role of ATP and mtROS in CD8⁺ T cell 3D migration, we performed additional experiments.

First, to modulate ATP levels without affecting mtROS, we incubated migrating cells with the uncoupling agent CCCP for 1 h. This led to a reduction in MMP (Supplementary Fig. 4B) and ATP-linked mitochondrial respiration (Supplementary Fig. 4C) without affecting mtROS level (Supplementary Fig. 4D). Under this condition, CD8⁺ T cell 3D motility was strongly inhibited (Supplementary Fig. 4E). To further confirm the correlation between ATP and 3D motility, we expressed the ATP biosensor PercevalHR[39,40] in CD8⁺ T cells and analyzed their motility in a collagen gel. Motile cells showed higher levels of PercevalHR signal (a proxy for ATP amount) compared with immotile cells (Supplementary Fig. 4F). These data confirm the role of mitochondrial ATP in supporting CD8⁺ T cell 3D motility.

Second, to investigate mtROS, we first tested the effect of ROS scavengers on 3D motility. Of note, both pan-ROS scavenger N-acetylcysteine (NAC) and mtROS-specific scavenger MitoTEMPO significantly reduced 3D motility at doses affecting mtROS amount (Fig. 4D, E), suggesting that mtROS may have a positive role in supporting CD8⁺ T cell 3D motility. We then wondered whether increasing mtROS amount could lead to an increase in 3D motility. For this purpose, we incubated CD8⁺ T cells with rotenone, an inhibitor of the ETC-I complex, during 3D migration in collagen gel. Surprisingly, at a dose capable of increasing mtROS amount (10 nM), rotenone inhibited CD8⁺ T cell 3D migration (Fig. 4F). However, we noticed that at this dose rotenone also inhibited basal and ATP-linked respiration (Fig. 4G), and thus ATP production, which is known to positively support CD8⁺ T cell 3D migration. Interestingly, the rotenone-induced reduction in cell migration was abrogated in presence of oligomycin whereas it was still observed (even at stronger level) with the ROS scavenger NAC (Fig. 4H, I). These data indicate that the rotenone-induced inhibition of 3D motility was mainly due to the ATP depletion and not to increased mtROS amount. Also, the stronger effect of rotenone-induced reduction in 3D migration in the presence of NAC (Fig. 4I) suggests that mtROS may play a positive role in counteracting the reduction in motility due to ATP depletion in this condition. We observed similar results with antimycin-A, an inhibitor of the ETC-III complex (Fig. 4J–M). In this case, the key role of ATP depletion was even more evident, since antimycin-A began to inhibit motility at 1 nM dose, while significantly affecting mtROS amount (at least when assessed by MitoSox staining) only at higher doses (Fig. 4J). Last, we tested the ETC-II complex inhibitor 3-nitropropionic acid (3-NP). Noteworthy, 3-NP increased both mtROS amount and 3D motility (Fig. 4N) without affecting mitochondrial respiration and ATP production (Fig. 4O) at 10 μM, whereas at higher doses (1 mM) mtROS amounts are further increased but motility and ATP production are decreased (Fig. 4N, O). Interestingly, the increase in cell motility due to 3-NP at 10 μM was abrogated by NAC treatment (indicating its dependence on mtROS), while the reduction observed at 1 mM was still observed in presence of NAC (when ROS are scavenged) and only disappeared upon oligomycin treatment (indicating its dependence on ATP depletion) (Fig. 4P). We

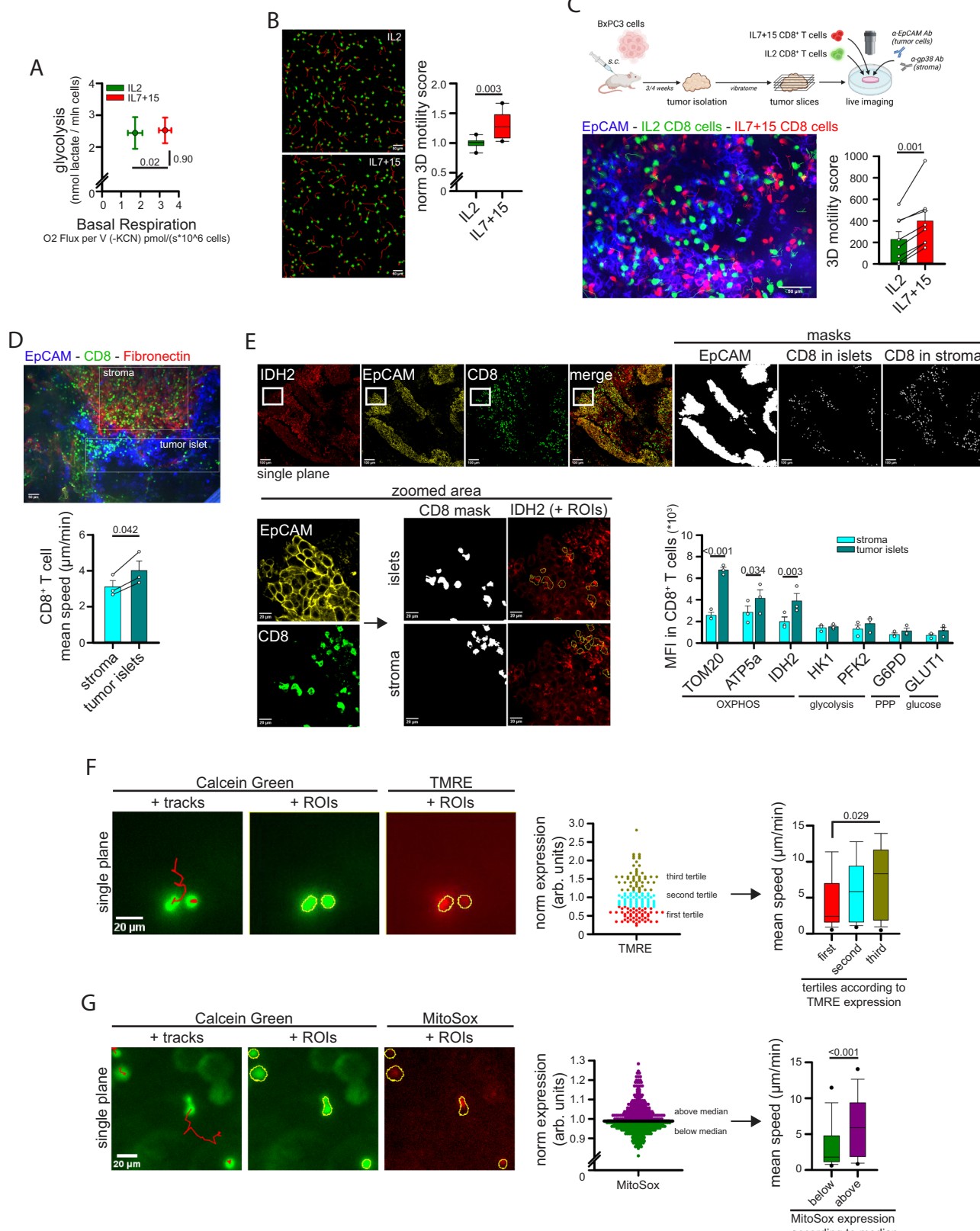

also confirmed these results using another ETC-II complex inhibitor, namely atpenin-A5 (Atp-A5). Also in this case, a low dose of Atp-A5 increased CD8⁺ T cell 3D migration in a NAC-dependent way, while at higher doses Atp-A5 reduced motility because of ATP depletion despite the concomitant increase in mtROS amount (Supplementary Fig. 4G, H).

Taken together, our data suggest that both mitochondrial ATP and mtROS support CD8⁺ T cell 3D motility. However, ATP levels play a prominent role, and mtROS can only further sustain motility if their accumulation does not affect concomitantly the OXPHOS-dependent ATP production.

**Fig. 3 | CD8⁺ T cell mitochondrial metabolism and 3D motility correlate across multiple contexts. A, B** CD8⁺ T cells have been activated (3d) and expanded (4d) in the presence of IL-2 or IL-7 + IL-15 (IL7 + 15). After 7 days, the energy plot (**A**, raw measurements are reported in Figure S3A, B, Linear Regression Test from mean values of S3A [$n = 5$ independent biological samples] and S3B [$n = 4$ independent biological samples]), and normalized 3D motility Score (**B**, $n = 8$ movies, unpaired two-tailed Student's T-Test) were calculated. **C** Experimental layout on top. 3D motility of activated Calcein Green-stained IL2 CD8⁺ T cells (green) and Calcein Red-stained IL7 + 15 CD8⁺ T cells (red) overlaid onto viable tumor slices (tumor cells in blue, EpCAM) derived from the BxPC3/NSG model ($n = 8$ slices, paired Student's T-Test). **D** 3D motility of resident CD8⁺ T cells (anti-CD8, green) in viable tumor slices (tumor cells in blue, EpCAM; stroma in red, fibronectin) derived from human NSCLC biopsies ($n = 3$ slices, paired Student's T-Test). **E** Mean level of expression (MFI) of the indicated proteins calculated in CD8⁺ T cells located in tumor islets or stroma of NSCLC biopsies, as identified from immunofluorescent staining (see

Methods for details) from $n = 3$ different blood donors (Two-Way ANOVA with posthoc Holm Sidak's). Confocal images on top and bottom left. The panel shows representative images of IDH2 staining only. **F, G** Activated CD8⁺ T cells have been stained with Calcein Green (green) and TMRE (**F**, red) or MitoSox (**G**, red) and 3D motility was assessed in collagen gel. For each cell, motility was evaluated by TrackMate analysis (tracks in red) and TMRE or MitoSox MFI was calculated at the first timepoint (cells were identified as ROIs using ImageJ). Cells were then stratified according to TMRE or MitoSox expression and motility was calculated in each subgroup (graphs on the right) (**F**, $n = 150$ cells, ANOVA on Ranks with posthoc Tukey's; **G**, $n = 420$ cells, MW Rank Sum Test). Data are expressed as mean ± SEM in (**A, C, D, E**), and as box plot (center line, median; box limits, upper and lower quartiles; whiskers, 1.5x interquartile range; points, 5th and 95th percentiles) in (**B, F, G**). Scale bar, 50 μm in (**B, D**), 20–100 μm in **E** and 20 μm in (**F, G**). Scheme in (**C**) has been created with BioRender.com.

Last, CD8⁺ T cell motility in a complex environment depends on actomyosin contractility and the ability to rearrange the cytoskeleton to support migration. We therefore wondered whether mitochondrial metabolism may impact on cytoskeleton rearrangements during CD8⁺ T cell migration into a collagen gel. Remarkably, we found that oligomycin treatment (which impacts both ATP and mtROS production at high doses, see Fig. 4B, C) significantly reduces the amount of polymerized actin (phalloidin staining) and tubulin (antibody staining in PHEM buffer) in migrating CD8⁺ T cells (Supplementary Fig. 4I). A reduction in the level of phosphorylated myosin light chain 2 (pMLC2) was also observed (Supplementary Fig. 4I). These data argue that mitochondrial metabolism is crucial for providing the energy for actomyosin contractility and cytoskeleton rearrangements required to promote cell migration in a 3D environment.

## Mitochondrial metabolism also sustains CD4⁺ T cell 3D motility

We wondered whether mitochondrial metabolism also plays a role in supporting CD4⁺ T cell 3D motility like what has been observed for CD8⁺ cells. To answer this question, naïve CD4⁺ T cells were activated and cultured like CD8⁺ T cells and their motility was assessed in a 3D collagen gel. We found that: (*i*) prolonged deprivation of glucose or glutamine impacted similarly on 3D motility (Supplementary Fig. 5A); (*ii*) 2DG and CB-839 inhibitors reduced similarly 3D motility, while etomoxir had no effect (Supplementary Fig. 5B); (*iii*) the strongest inhibition of 3D motility was observed in presence of 6,8bOA (Supplementary Fig. 5B); (*iv*) complete blockade of TCA cycle using 6,8bOA, CB-839 and etomoxir completely abrogated 3D motility (Supplementary Fig. 5B); and (*v*) both ATP and mtROS production sustained 3D motility (Supplementary Fig. 5C, D).

Overall, these data suggest that glucose- and glutamine-fueled TCA cycle generating OXPHOS-derived ATP and mtROS is the main metabolic pathway sustaining 3D motility also in CD4⁺ T cells.

## Pharmacological approaches promoting mitochondrial metabolism in human CD8⁺ T cells increase 3D intratumoral motility

We then tested strategies promoting mitochondrial metabolism for their ability to enhance CD8⁺ T cell 3D migration. To this aim, we first activated and cultured CD8⁺ T cells in the presence of rapamycin (Rapa-CD8 cells), an mTOR inhibitor known to sustain mitochondrial metabolism in immune cells[41,42] (see scheme in Fig. 5A). One-week treatment with rapamycin did not affect viability (Supplementary Fig. 6A) but significantly increased mitochondrial mass, as assessed by quantification of mtDNA amount (Fig. 5B), mitochondrial mass (Supplementary Fig. 6B) and levels of ETC complexes (Supplementary Fig. 6C). Consequently, Rapa-CD8 cells showed a higher maximal respiratory capacity (Fig. 5C), while reducing glycolysis (Supplementary Fig. 6D). Interestingly, these cells also showed an improved 3D motility in collagen gel (Fig. 5D and Supplementary Movies 36, 37), an effect dependent on the increased mitochondrial activity, since both

oligomycin (Fig. 5E and Supplementary Movies 38–41) and hypoxia (Supplementary Fig. 6E) abrogated it. Mechanistically, we found that such rapamycin-dependent enhancement of mitochondrial activity also increased the phosphorylation level of myosin light chain 2 (pMLC2) (Supplementary Fig. 6F), thus sustaining actomyosin contractility to support active migration. Remarkably, in contrast to what was observed for ctrl-CD8 cells, etomoxir significantly reduced the 3D motility of Rapa-CD8 cells (Fig. 5F), suggesting that rapamycin opened FAs usage to sustain motility. Rapa-CD8 cells also showed better intratumoral motility compared with ctrl-CD8 cells, as observed using tumor slices derived from the BxPC3-NSG model (Fig. 5G and Supplementary Movie 42). Moreover, to monitor 3D lymphocyte infiltration into a solid environment, CD8⁺ T cells were added adjacent to a BxPC3 tumor cells-filled collagen gel and their ability to infiltrate was evaluated. Remarkably, Rapa-CD8 cells infiltrated the 3D environment faster than ctrl-CD8 cells (Fig. 5H). Consistent with previous studies[43,44], rapamycin treatment also increased CD8⁺ T cell differentiation towards a more primitive (Tscm-like) state (Supplementary Fig. 6G) and reduced IFNγ and perforin production (but not granzyme B and TNF) upon restimulation (Supplementary Fig. 6H), presumably due to inhibition of glycolysis[45].

In vitro treatment with rapamycin could therefore be an effective strategy for enhancing the infiltration and intratumoral motility of CD8⁺ T cells. To ensure that such an effect could be still maintained several days after rapamycin withdrawal, CD8⁺ T cells treated in vitro for 10 days were deprived of rapamycin and their functions were evaluated some days after (see scheme in Supplementary Fig. 6I). Of note, 4/7 days after rapamycin withdrawal, cytokine production was not only restored but even increased in Rapa-CD8 cells (Supplementary Fig. 6J), and these cells retained a better 3D motility compared with ctrl-CD8 cells (Supplementary Fig. 6K). Moreover, although rapamycin treatment reduced in vitro CD8⁺ T cell expansion (Supplementary Fig. 6L), this effect was mainly limited to the first few days of culture and disappeared after rapamycin removal (Supplementary Fig. 6M), suggesting that expansion of Rapa-CD8 cells in the host may be similar to control cells. Last, Rapa-CD8 cells also showed an enhanced 3D motility in a different type of collagen (telo-collagen) compared to ctrl-CD8 cells (Supplementary Fig. 6N).

Next, we activated and cultured CD8⁺ T cells in the presence of bezafibrate (Beza-CD8 cells), which can support mitochondrial metabolism in T cells[46] (see scheme in Fig. 5I). Bezafibrate treatment did not affect T cell viability (Supplementary Fig. 7A) but increased mitochondrial respiration (Fig. 5J), although this effect was not associated to an increase in mitochondrial mass (Supplementary Fig. 7B). Of note, Beza-CD8 cells showed an oligomycin-dependent increase in 3D motility (Fig. 5K and Supplementary Movies 43–46). Unlike Rapa-CD8 cells, Beza-CD8 cells did not acquire the ability to use FAs to fuel their motility (Fig. 5L), and their glycolysis and in vitro proliferation were not

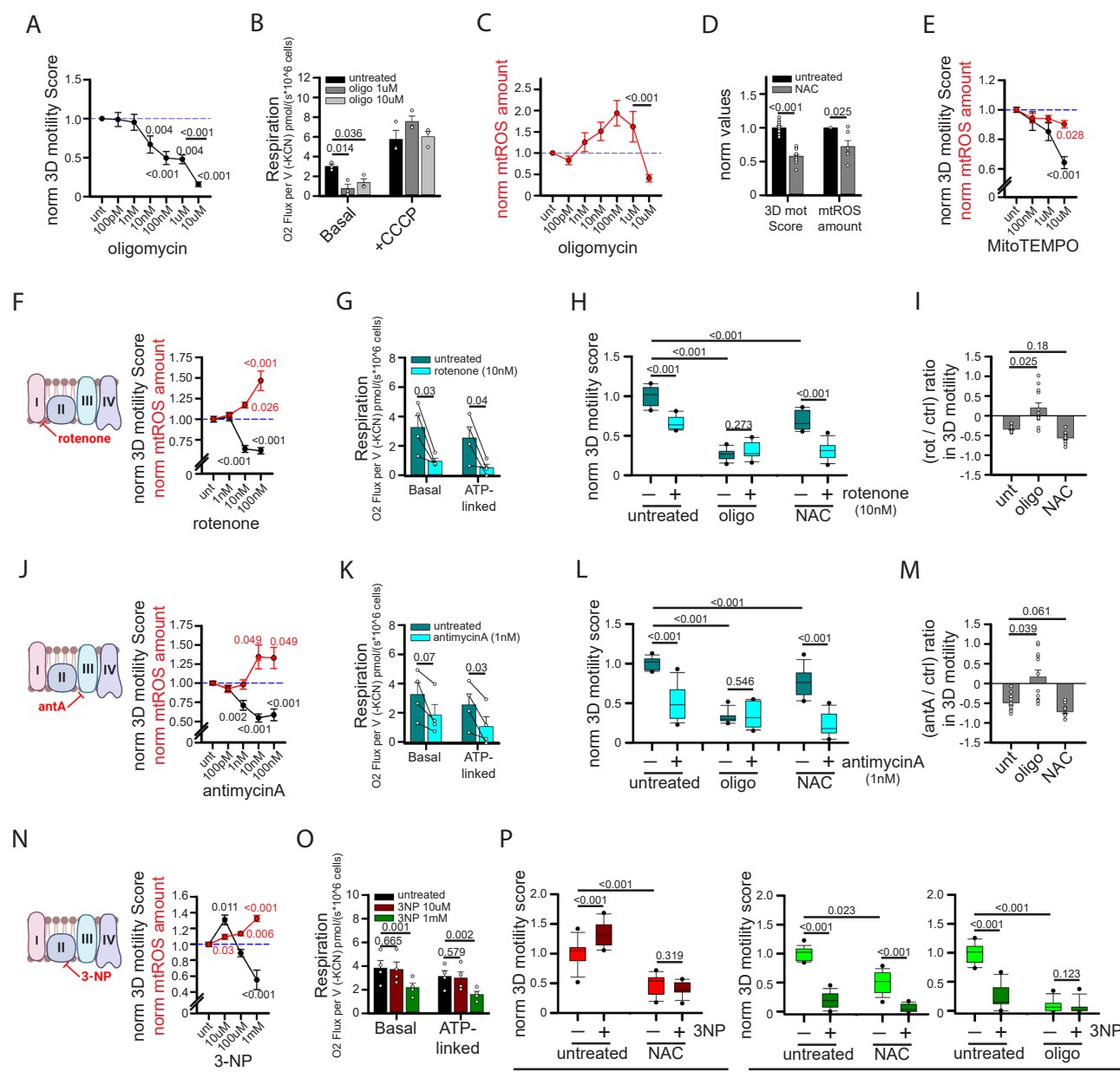

**Fig. 4 | Mitochondria sustain CD8⁺ T cell 3D motility by providing both ATP and mtROS. A–C** Normalized 3D motility Score (**A**, $n = 8$ movies, ANOVA on Ranks with posthoc Student–Newman–Keuls's), basal and CCCP-induced respiration (**B**, $n = 3$ independent biological samples, One-Way ANOVA with posthoc Holm Sidak's) and normalized mtROS amount (MitoSox/FSC ratio; **C**, $n = 7$ independent biological samples, ANOVA on Ranks with posthoc Student–Newman–Keuls's) of activated CD8⁺ T cells in the presence of the indicated doses of oligomycin. **D**, **E** Normalized 3D motility Score (**D**, $n = 12$ movies, unpaired two-tailed Student's T-Test; **E**, $n = 15$ movies, ANOVA on Ranks with posthoc Tukey's) and mtROS amount (MitoSox/FSC ratio; **D**, $n = 6$ independent biological samples, unpaired two-tailed Student's T-Test; **E**, $n = 6$ independent biological samples, ANOVA on Ranks with posthoc Tukey's) of activated CD8⁺ T cells in the presence of N-acetylcysteine (NAC) (in **D**) or MitoTEMPO (in **E**). **F–P** Normalized 3D motility Score (**F**, $n = 16$ movies, ANOVA in Ranks with posthoc Dunnet's; **J**, $n = 16$ movies, ANOVA in Ranks with posthoc Dunnet's; **N**, $n = 9$ movies, One-Way ANOVA with posthoc Holm Sidak's vs ctrl), mtROS amount (MitoSox/FSC ratio; **F**, **J**, $n = 7$ independent biological samples, ANOVA in Ranks with posthoc Dunnet's; **N**, $n = 6$ independent biological samples, One-Way ANOVA with posthoc Holm Sidak's vs ctrl) and basal and ATP-linked respiration (**G**, **K**, $n = 4$ independent biological

samples, unpaired Student's T-Test; **O**, $n = 4$ independent biological samples, One-Way ANOVA on Repeated Measurements with posthoc Holm Sidak's) of activated CD8⁺ T cells in the presence of the indicated doses of rotenone (**F–K**), antimycinA (antA) (**J–M**) or 3-nitropropionic acid (3-NP) (**N–P**). In (**H**, **L**, **P**), 3D motility was evaluated also in the presence of N-acetylcysteine (NAC) or oligomycin (oligo) (**H–L**, $n = 12$ movies, Two-Way ANOVA with posthoc Holm Sidak's; P, 3-NP 10 µM, $n = 12$ movies, Two-Way ANOVA with posthoc Holm Sidak's; oligo + 3-NP 1 mM, $n = 12$ movies, ANOVA on Ranks with posthoc Tukey's; NAC + 3-NP 1 mM, $n = 18$ movies, ANOVA on Ranks with posthoc Tukey's) and the relative (rotenone/control) or (antimycinA/control) changes in 3D motility Score for each condition were reported in (**I**, **M**) (for both $n = 12$ movies, ANOVA on Ranks with posthoc Dunnet's). Schemes on the left in (**F**, **J**, **N**) indicate ETC complexes inhibited by each drug. Please note that the same controls were used for data in Figs. 4G and 4K since performed in the same experiments (data were split to improve clarity of the presentation). Data are expressed as mean ± SEM in (**A–G**, **J**, **K**, **N**, **O**), and as box plot (center line, median; box limits, upper and lower quartiles; whiskers, 1.5x interquartile range; points, 5th and 95th percentiles) in (**H**, **I**, **L**, **M**, **P**). Schemes in (**F**, **J**, **N**) have been created with BioRender.com.

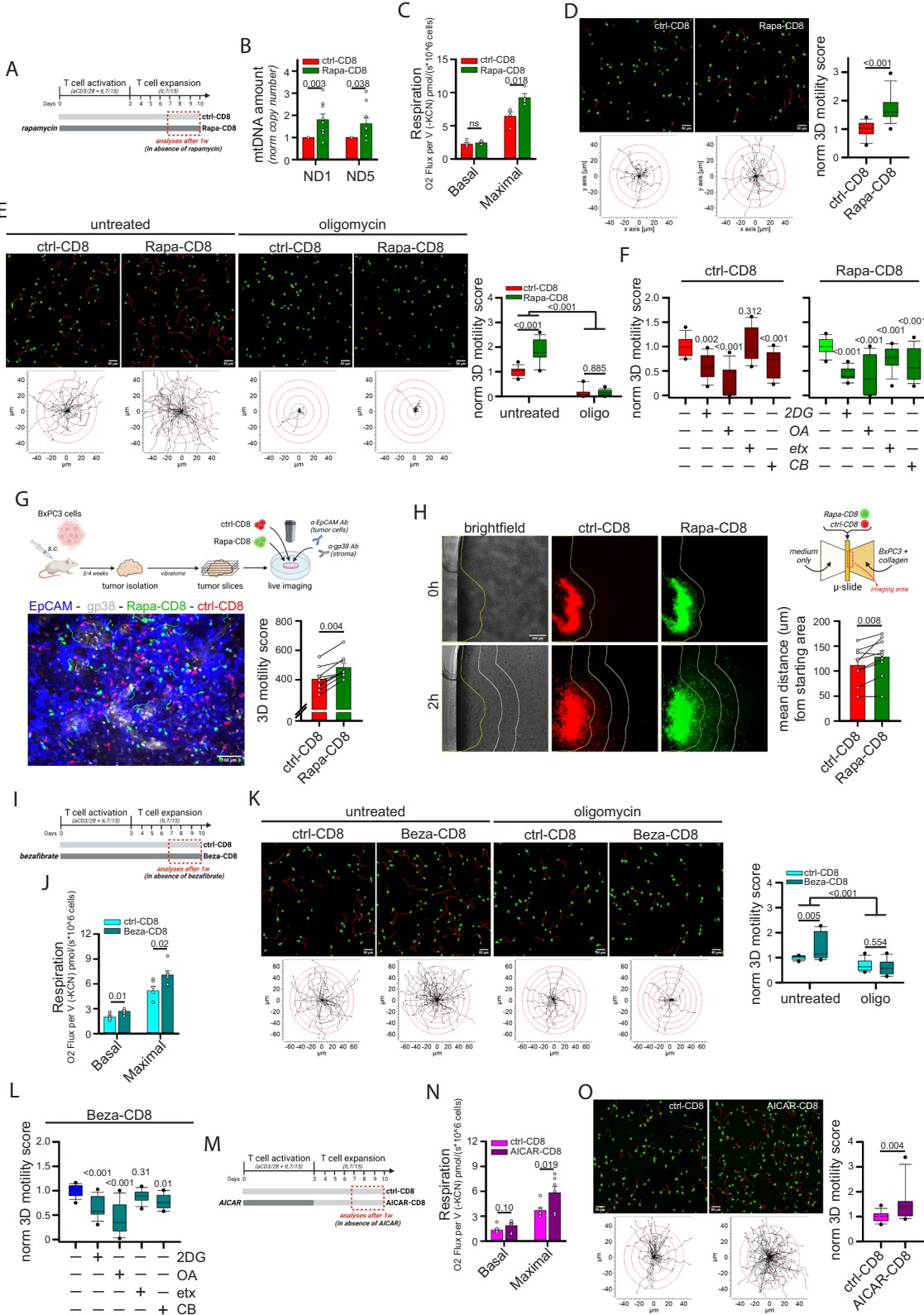

affected (Supplementary Fig. 7C, D). The enhanced 3D motility was also observed using telo-collagen gel (Supplementary Fig. 7E).

Third, we activated CD8[+] T cells in the presence of AICAR, an activator of the energy regulator AMPK[47] (AICAR-CD8 cells), for 3 days before expanding them in vitro in the absence of the drug (a more prolonged treatment was toxic) (see scheme in Fig. 5M). Like bezafibrate, AICAR had no impact on cell viability (Supplementary Fig. 7F),

mitochondrial mass (Supplementary Fig. 7G) or glycolysis (Supplementary Fig. 7H) but significantly increased maximal respiration (Fig. 5N) and 3D motility (Fig. 5O and Supplementary Movies 47, 48) in CD8[+] T cells. Proliferation was reduced similarly to what was observed for rapamycin (Supplementary Fig. 7I).

Last, βNAD treatment (Supplementary Fig. 7J), which modulates mitochondrial metabolism in T cells[48], increased ATP-linked

**Fig. 5 | Drugs promoting mitochondrial metabolism increase CD8⁺ T cell 3D motility. A–F** Experimental layout in A. CD8⁺ T cells have been cultured in ctrl condition (ctrl-CD8) or in the presence of rapamycin (Rapa-CD8). After 1w, rapamycin was removed the day of analysis and the following parameters were measured: mtDNA amount (normalized $2^{-\Delta Ct}$ values between mtDNA genes ND1 or ND5 and nuclear gene 18 S; **B**, $n = 9$ independent biological samples, MW Rank Sum Test), basal and maximal respiration (**C**, $n = 4$ independent biological samples, unpaired two-tailed Student's T-Test), normalized 3D motility Score in the presence or not of oligomycin (**D**, $n = 30$ movies, MW Rank Sum Test; **E**, $n = 15$ movies, ANOVA on Ranks with posthoc Student–Newman–Keuls's). In (**F**), the relative 3D motility Score is reported after normalizing to 1 the mean score for both ctrl-CD8 and Rapa-CD8 cells (2DG: 2-deoxy-glucose, 6,8bOA: 6,8-bis(benzylthio)octanoic acid; etx: etomoxir; CB: CB-839) (ctrl-CD8, $n = 12$ movies and Rapa-CD8, $n = 15$ movies, ANOVA on Ranks with posthoc Student–Newman–Keuls's). **G** Experimental layout on top. 3D motility of Calcein Green-stained ctrl-CD8 cells (shown in red for consistency) and Calcein Red-stained Rapa-CD8 cells (shown in green for consistency) overlaid onto viable tumor slices derived from the BxPC3/NSG model (tumor cells in blue, EpCAM; stroma in gray, gp38) ($n = 9$ slices, Wilcoxon paired Rank Sum Test). **H** Infiltration of Calcein Red-stained ctrl-CD8 cells (red) and Calcein Green-stained Rapa-CD8 cells (green) into a collagen solution containing human BxPC3 tumor cells (scheme on the top right, see Methods for details). Yellow lines identify starting position of cells at 0 h. White lines represent 200 μm

increments. The mean distance from starting area for the two populations is reported in the graph ($n = 9$ slides, paired Student's T-Test). **I–L** Experimental layout in (**I**). CD8⁺ T cells have been cultured in ctrl condition (ctrl-CD8) or in the presence of bezafibrate (Beza-CD8). After 1w, bezafibrate was removed and the following parameters were measured: basal and maximal respiration (**J**, $n = 6$ independent biological samples, unpaired two-tailed Student's T-Test) and normalized 3D motility Score in the presence or not of oligomycin (**K**, $n = 12$ movies, ANOVA on Ranks with posthoc Student–Newman–Keuls's). In (**L**) ($n = 15$ movies, ANOVA on Ranks with posthoc Dunnet's), the relative 3D motility Score is reported after normalizing to 1 the mean score for untreated Beza-CD8 cells (as in **F**). **M–O** Experimental layout in (**M**). Activated CD8⁺ T cells have been cultured in ctrl condition (ctrl-CD8) or in the presence of AICAR (AICAR-CD8). After 3d, AICAR was removed, and the cells expanded in the control condition. The following parameters were measured starting from 7d: basal and maximal respiration (**N**, $n = 7$ independent biological samples, unpaired two-tailed Student's T-Test) and normalized 3D motility Score (**O**, $n = 15$ movies, MW Rank Sum Test). Data are expressed as mean ± SEM in (**B, C, G, H, J, N**), and as box plot (center line, median; box limits, upper and lower quartiles; whiskers, 1.5x interquartile range; points, 5th and 95th percentiles) in (**D–F, K, L, O**). Scale bar, 50 μm in (**D, E, K, O**), 100 μm in (**G**) and 200 μm in (**H**). Schemes in (**A, G, H, I, M**) have been created with BioRender.com.

respiration (Supplementary Fig. 7K) and 3D motility (Supplementary Fig. 7L) in CD8⁺ T cells. Remarkably, the effects of rapamycin, AICAR and βNAD were due to a metabolic reprogramming since administration of these drugs during the motility assay had no effect or even inhibited 3D motility (Supplementary Fig. 7M).

Overall, these data support the notion that CD8⁺ T cell 3D intratumoral motility and infiltration can be enhanced using pharmacological strategies targeting mitochondrial metabolism.

## Rapamycin-treated human CD8⁺ CAR T cells exhibit superior mitochondrial metabolism and 3D intratumoral motility

The ability to equip CD8⁺ CAR T cells with better infiltration and intratumoral motility could significantly improve the efficacy of CAR-based immunotherapies against solid cancers. Therefore, we first decided to investigate if mitochondrial metabolism is also the key driver of 3D motility in CD8⁺ CAR T cells. To this aim, CD8⁺ T cells were transduced with lentiviral particles to generate anti-EGFR CD8⁺ CAR T cells and their 3D motility was assessed in collagen gels. Consistent with what was observed in normal T cells, glucose and glutamine similarly supported CD8⁺ CAR T cells 3D motility (Supplementary Fig. 8A) and TCA-cycle inhibitor 6,8bOA strongly reduced it (Supplementary Fig. 8B). Also, 2DG and CB-839 had a similar effect, while etomoxir had no impact (Fig. S8B). Motility was also strongly suppressed by oligomycin (Supplementary Fig. 8C). Finally, CRISPR/Cas9-mediated double editing of *PDHA1* and *OGDH* genes (Supplementary Fig. 8D) led to a significant decrease in 3D motility (Supplementary Fig. 8E). Overall, these data indicate that mitochondrial metabolism strongly supports also CD8⁺ CAR T cell 3D motility.

Next, we tested whether pharmacological strategies enhancing the mitochondrial activity of CD8⁺ CAR T cells could also increase intratumoral motility and infiltration. Among the different drugs tested, we selected rapamycin since this drug produced the strongest effects in terms of motility and increase in mitochondrial mass. CD8⁺ T cells were activated in presence of rapamycin, transduced with anti-EGFR CAR lentiviral particles (Rapa-CD8^CAR cells), and further expanded for up to 10 days in the presence of the drug, which was removed on the day of the analysis (see scheme in Fig. 6A). Rapamycin treatment had no impact on CAR expression (Supplementary Fig. 9A) but significantly increased maximal respiration (Fig. 6B), while reducing glycolysis (Supplementary Fig. 9B). Remarkably, Rapa-CD8^CAR cells showed enhanced 3D motility in collagen gel (Fig. 6C and Supplementary Movies 49, 50), an effect mainly due to a higher mean speed of moving cells (Supplementary Fig. 9C). To test the motility of these

cells in a solid TME model, NSG mice were i.v. inoculated with human A549 lung tumor cells (see scheme in Fig. 6D), which rapidly populated the lungs generating a relevant TME structure composed of medium-sized tumor islets surrounded by a large stroma. After 3 weeks, lungs were excised and cut into viable slices. Interestingly, Rapa-CD8^CAR cells showed improved intratumoral motility when overlaid onto the lung tumor slices compared to ctrl-CD8^CAR cells (Fig. 6D and Supplementary Movies 51, 52). We also evaluated the ability of CD8⁺ CAR T cells to migrate across a TNF-activated HUVEC monolayer in a transwell assay as a model of T cell extravasation. Rapa-CD8^CAR cells migrated through the HUVEC monolayer faster than control cells (Fig. 6E).

Besides migration, a functional CAR T cell must display additional activities, such as persistence in the host, cytotoxicity, and cytokine production. Therefore, we tested whether these functions were affected by rapamycin treatment. Consistent with what was observed for Rapa-CD8 cells, Rapa-CD8^CAR cells showed a more primitive (Tscm-like) state (Fig. 6F), which could presumably be responsible for the reduced in vitro expansion rate of these cells (Supplementary Fig. 9D). Cytotoxicity and cytokine production were assessed 1-week after rapamycin removal (see scheme in Fig. 6A), as we reasoned that Rapa-CD8^CAR cells should only fulfill these functions once in the TME and therefore several days after injection into patients (and rapamycin removal). Compared with ctrl-CD8^CAR, Rapa-CD8^CAR cells showed increased production of IFNγ and TNF cytokines upon restimulation (Fig. 6G, H) and slightly reduced cytotoxicity against human A549 (EGFR⁺) lung tumor cells but only at a low CAR:target ratio (Fig. 6I).

In summary, rapamycin treatment generates functional CD8⁺ CAR T cells empowered with superior infiltration ability and intratumoral motility.

## Rapamycin-treated human CD8⁺ CAR T cells are more effective at infiltrating solid tumor masses in preclinical solid tumor xenograft models

We then decided to test whether Rapa-CD8^CAR cells showed enhanced infiltration into solid tumor islets in vivo. To this aim, we chose two preclinical models based on the injection of human A549 lung tumor cells (EGFR⁺) into NSG mice. First, A549 cells were i.v. injected to generate lung tumors (orthotopic model, see scheme in Fig. 7A). After 3 weeks, mice were i.v. inoculated with anti-EGFR ctrl- or Rapa-CD8^CAR cells. To evaluate CAR T cell ability to infiltrate tumor islets, mice were sacrificed 4 days after, and lungs were analyzed. The total amount of ctrl- or Rapa-CD8^CAR cells infiltrating the lungs was similar (Fig. 7B and Supplementary Fig. 10A), consistent with the notion that the lungs are

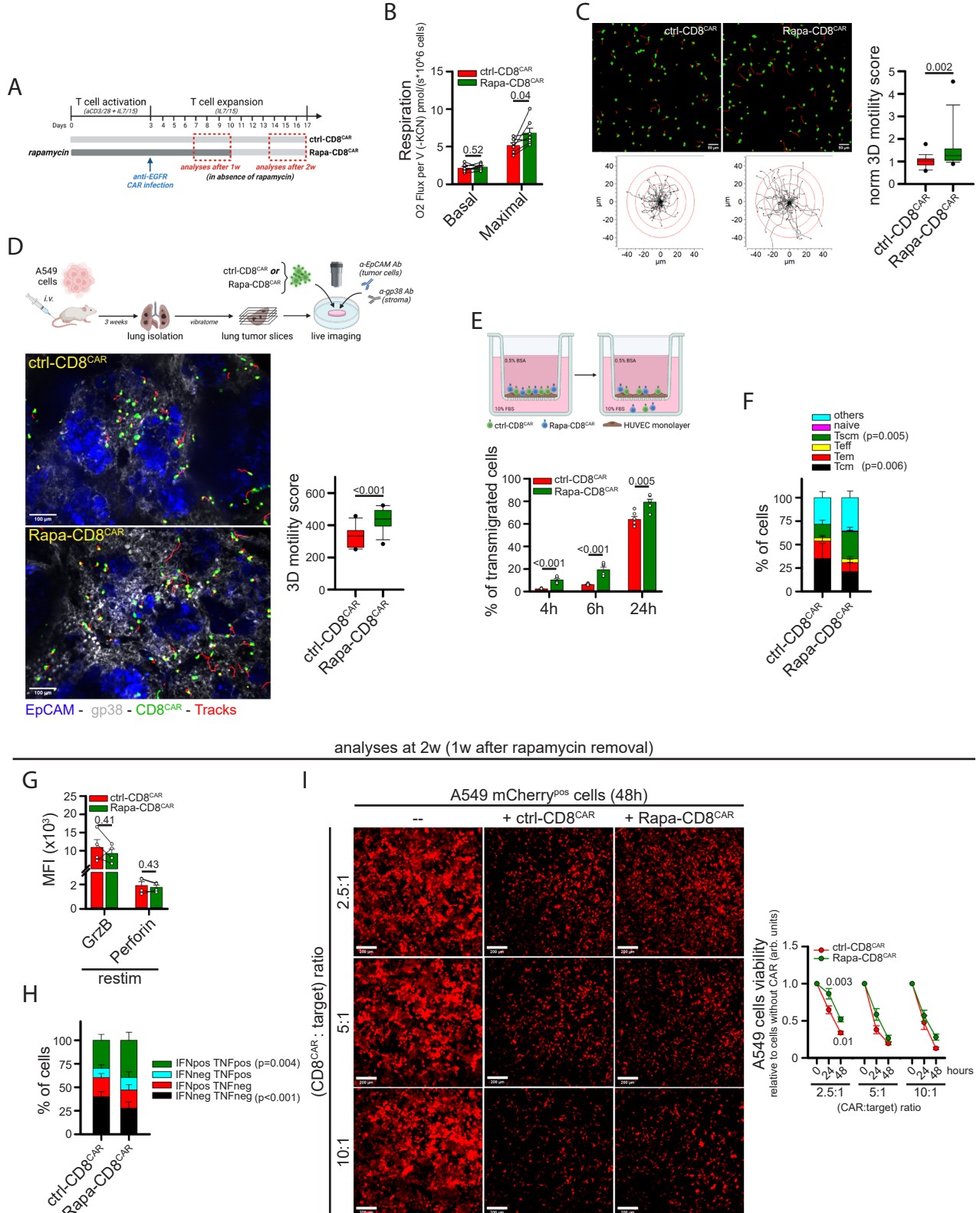

a particularly favorable site for T cell infiltration[49]. However, we observed that Rapa-CD8[CAR] cells were significantly more accumulated in tumor islets compared with ctrl-CD8[CAR] cells, which remained largely in the surrounding stroma (Fig. 7B). The better infiltration of Rapa-CD8[CAR] into tumor islets was also confirmed by a competition assay upon injection of eFluor670 (eF670)-labelled ctrl-CD8[CAR] and CFSE-labelled Rapa-CD8[CAR] cells mixed in a 1:1 ratio into A549-derived lung

tumor-bearing NSG mice (Supplementary Fig. 10B). Of note, such an increased infiltration into tumor islets of Rapa-CD8[CAR] cells was associated with higher levels of active-caspase-3 (a marker of apoptosis) in tumor islets (Fig. 7C), indicating a stronger anti-tumor response. We also assessed the ability of Rapa-CD8[CAR] cells to reduce tumor growth in this model by injecting suboptimal amount of CAR T cells into NSG mice bearing a lung tumor derived from the injection of Luciferase[pos]

**Fig. 6 | Rapamycin treatment promotes intra-tumoral motility and transmigration in anti-EGFR CD8⁺ CAR T cells. A** Experimental layout. Anti-EGFR CD8⁺ CAR T cells were cultured in a control medium (ctrl-CD8^CAR) or in the presence of rapamycin (Rapa-CD8^CAR). All assays were performed in the absence of rapamycin. **B, C** Measurements of basal and maximal respiration (**B**, $n = 7$ independent biological samples, unpaired two-tailed Student's t-Test) and normalized 3D motility Score (**C**, $n = 18$ movies, MW Rank Sum Test). **D** Experimental layout on top. 3D motility of activated Calcein Green-stained ctrl-CD8^CAR and Rapa-CD8^CAR cells (cells in green, tracks in red) overlaid onto viable lung tumor slices (tumor islets in blue, EpCAM; stroma in gray, gp38) derived from i.v. injection of A549 lung tumor cells into NSG mice ($n = 12$ slices, unpaired two-tailed Student's t-Test). **E** Percentage of Calcein Green-stained ctrl-CD8^CAR and eFluo670-stained Rapa-CD8^CAR cells (mixed in a 1:1 ratio) that transmigrated across a TNF-activated HUVEC monolayer in a transwell system ($n = 6$ independent biological samples, Two-Way ANOVA with posthoc Holm Sidak's). **F** Relative proportion of indicated T cell subsets in ctrl-CD8^CAR and Rapa-CD8^CAR cells ($n = 4$ independent biological samples, Two-Way ANOVA on repeated Measurements with posthoc Holm Sidak's). **G, H** At 2 weeks (1 week after rapamycin removal) cells were restimulated for 5 h with human TransAct + BfdA. GranzymeB (GrzB) and Perforin levels (MFI) are indicated in **G** ($n = 4$ independent biological samples, unpaired two-tailed Student's t-test). For IFNγ and TNF, the percentage of cells producing these cytokines is indicated in (**H**) ($n = 4$ independent biological samples, Two-Way ANOVA on Repeated Measurements with posthoc Holm Sidak's). **I** Cytotoxicity of ctrl-CD8^CAR and Rapa-CD8^CAR cells against A549 (EGFR⁺ mCherry⁺) tumor cells 1 week after rapamycin removal. A549 cell viability was evaluated by measuring mCherry fluorescence (MFI) for each microscope field at the indicated time points ($n = 12$ microscope fields, Two-Way ANOVA with posthoc Holm Sidak's for 2.5:1 and 10:1 ratio; ANOVA on Ranks with posthoc Tukey's for 5:1 ratio). Data are expressed as mean ± SEM in (**B, E–I**), and as box plot (center line, median; box limits, upper and lower quartiles; whiskers, 1.5x interquartile range; points, 5th and 95th percentiles) in (**C, D**). Scale bar, 50 μm in (**C**), 100 μm in D and 200 μm in (**I**). Schemes in (**A, D, E**) have been created with BioRender.com.

A549 cells 3 days before (see scheme in Fig. 7D). Interestingly, only mice inoculated with Rapa-CD8^CAR cells, but not with ctrl-CD8^CAR, showed a significant reduction in tumor growth compared with untreated mice (Fig. 7D).

To evaluate the effectiveness of Rapa-CD8^CAR cells in a different TME, A549 cells were s.c. inoculated into NSG mice to generate a subcutaneous tumor (see scheme in Fig. 7E). The morphology of TME is different in this model compared with the orthotopic one (tumor cells do not accumulate in stroma-surrounded islets but in large areas with fewer adjacent stroma) and vasculature is more deregulated (as frequently observed in human solid tumors[50]), making CAR T cell infiltration more difficult compared with lungs[49]. Mice were i.v. inoculated with anti-EGFR ctrl- or Rapa-CD8^CAR cells 5/6 weeks after inoculation of tumor cells, and then sacrificed 4 days after. Remarkably, despite no alterations in the amount of tumor cells per section (Supplementary Fig. 10C), Rapa-CD8^CAR cells showed increased infiltration into the tumor mass, as indicated by an increased amount of CAR T cells per section (Fig. 7E), and also accumulated significantly more in tumor islets compared to ctrl-CD8^CAR cells, which remained more confined in the few stromal areas (Fig. 7E). The increased infiltration of Rapa-CD8^CAR into the tumor mass was also confirmed by a competition assay upon injection of eFluor670 (eF670)-labelled ctrl-CD8^CAR and CFSE-labelled Rapa-CD8^CAR cells mixed in a 1:1 ratio into A549-derived subcutaneous tumor-bearing NSG mice, as shown by flow cytometry and immunofluorescence (Supplementary Fig. 10D). To evaluate anti-tumor activity in this model, CD8⁺ CAR T cells were infused in tumor-bearing NSG mice 10 days after s.c. injection of A549 cells and tumor growth was followed for 1 month (see scheme in Fig. 7F). Noteworthy, while injection of ctrl-CD8^CAR cells had no effect, Rapa-CD8^CAR cells significantly reduced tumor growth compared to untreated mice (Fig. 7F, see also Supplementary Fig. 10E, F for individual growth curves and relative tumor growth of treated mice normalized over the growth of untreated mice in each cage to correct for potential cage effect). Last, we isolated tumor-infiltrating CD8⁺ CAR T cells at 38 days (i.e., 1 month after CAR T cell injection) and we found an increased although not significant ($p = 0.06$) number of Rapa-CD8^CAR cells in s.c. tumors compared with ctrl-CD8^CAR cells (Fig. 7G). We speculate that this increase may be associated with the more primitive (Tscm-like) initial differentiation state of Rapa-CD8^CAR cells (see Fig. 6F), which has been previously associated with improved CAR T cell long-term persistence[51–53].

Last, we excluded that the increased infiltration of Rapa-CD8^CAR cells into our tumor models was due to altered expression of cell adhesion molecules (CD62L, CD49d, CD2, CD44, CD103, CD11a, LPAM-1) and chemokine receptors (CCR5, CXCR3, CXCR4, CCR7, CCR3), whose levels were not altered compared to ctrl-CD8^CAR cells (Supplementary Fig. 10G), reinforcing the importance of the metabolic reprogramming induced by rapamycin to promote CAR T cell infiltration.

Taken together, our data demonstrate that in vitro rapamycin treatment generates CD8⁺ CAR T cells with enhanced ability to infiltrate into tumor islets, leading to a better control of tumor growth in two preclinical solid tumor models.

## Activation at 39 °C improves mitochondrial metabolism and intratumoral motility in CD8⁺ CAR T cells

To reinforce the idea that manipulation of mitochondrial metabolism increases CD8⁺ CAR T cell 3D motility, we used an alternative drug-free system by taking advantage of a recent finding describing that culturing CD8⁺ T cells under fever-like conditions (i.e., 39 °C) increases mitochondrial activity[54]. CD8⁺ T cells were activated for 3 days at either 37 °C (act-37 °C) or at 39 °C (act-39 °C) and then transferred to 37 °C for viral infection (anti-EGFR CAR) and subsequent expansion (see scheme in Fig. 8A). T cell activation performed at 39 °C had no impact on the level of CAR expression (Supplementary Fig. 11A) and cell viability (Supplementary Fig. 11B) but increased maximal mitochondrial respiration (Fig. 8B) without affecting mitochondrial mass, MMP, mtROS amount (Supplementary Fig. 11C), glycolysis rate (Fig. 8C), in vitro expansion (Fig. 8D) and cytokine production (Supplementary Fig. 11D). Remarkably, CD8⁺ CAR T cells activated at 39 °C showed (i) increased 3D motility in collagen gel (Fig. 8E and Supplementary Movies 53-56), an effect likely promoted by the increased mitochondrial activity since oligomycin abrogated it (Fig. 8E and Movies S53–56), and (ii) superior intratumoral motility when deposed onto viable tumor slices derived from the BxPC3-NSG model (Fig. 8F and Supplementary Movies 57, 58).

Overall, these data demonstrate that cell activation at 39 °C can be an efficient strategy to increase mitochondrial activity and hence intratumoral motility of CD8⁺ CAR T cells.

## Discussion

The ability of T cells to migrate and reach tumor cells determines the efficacy of T cell-based immunotherapy. Thus, there is a lot of interest in elucidating the mechanisms responsible for the migration of CD8 T cells in tumors and in developing strategies to boost this function. Studies performed using imaging techniques have underlined the importance of both environmental and cell-intrinsic factors in this process. In the present work, we identified the metabolic requirements of CD8⁺ T cell migration in a 3D environment (especially intra-tumoral one). Our results obtained in vitro and confirmed in ex vivo tumor slices support a key role of the TCA cycle fueled by glucose and glutamine, but no FAs, with glycolysis playing only a much minor role (Fig. S12). Strategies targeting mitochondrial metabolism were then developed and tested in several preclinical models including two xenograft mouse tumor models. We demonstrated that CD8⁺ CAR T cells with enhanced mitochondrial activity showed superior infiltration into solid tumors leading to better inhibition of tumor growth (Supplementary Fig. 12).

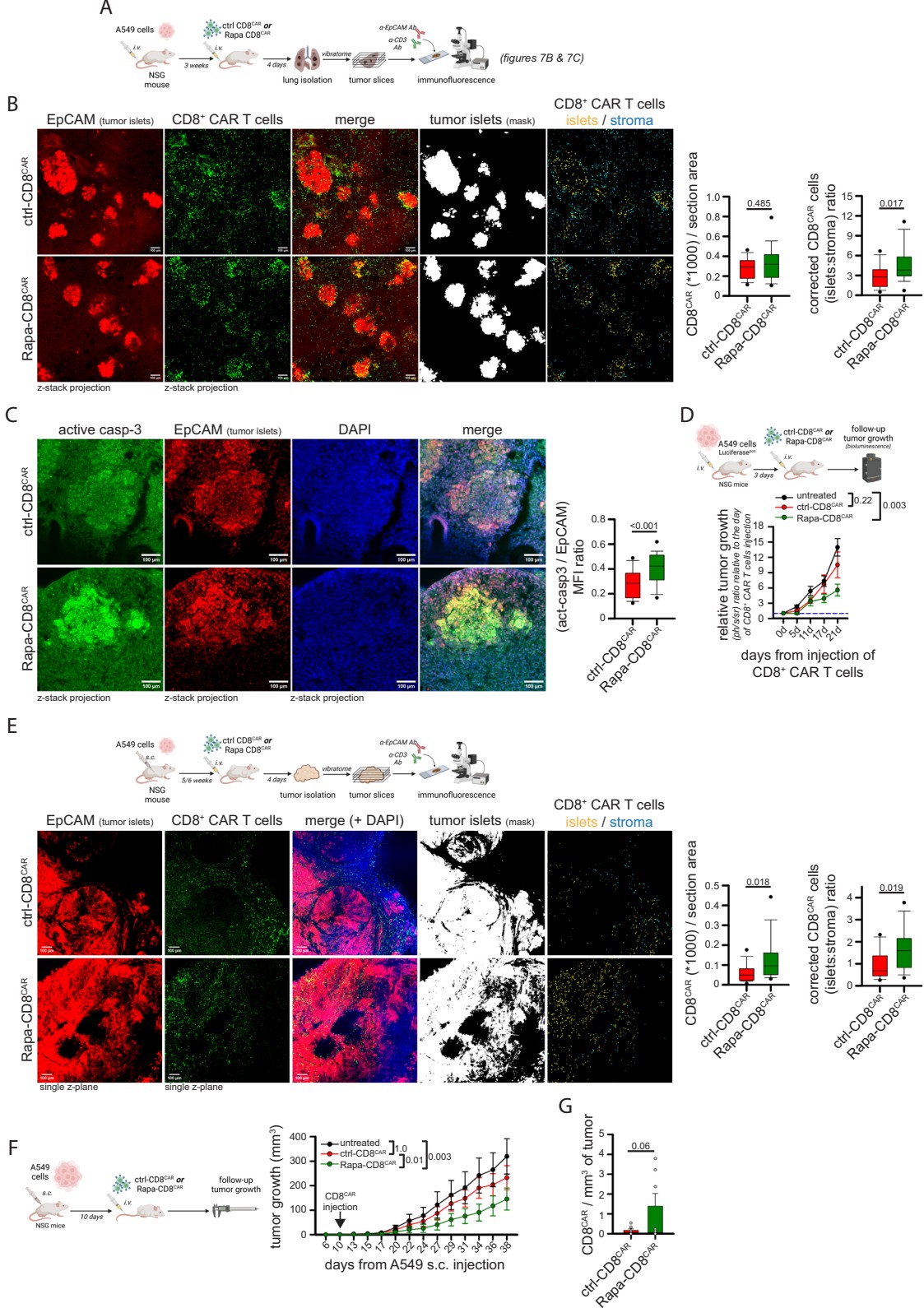

The poor role of glycolysis in supporting the amoeboid-like of CD8+ T cells in 3D environments contrasts clearly with what has been observed for migrating cancer cells. Here, glycolysis plays a much more important role by providing ATP locally at lamellipodia to sustain the formation of protrusion[55,56]. Moreover, mtROS have been descri-bed to positively support cancer cell migration and metastatic spread[57]. Interestingly, rotenone treatment has been shown to

promote cancer cell dissemination at doses that would inhibit OXPHOS-derived ATP generation[57], further reinforcing the role of glycolysis for energy production in these migrating cells. We also observed a positive role for mtROS in CD8+ T cell 3D motility. However, given the prominent role of mitochondrial metabolism and not gly-colysis, a subtle balance between mtROS and mitochondrial ATP (mito-ATP) production is required in CD8+ T cells to support 3D motility,

**Fig. 7 | Rapamycin-treated anti-EGFR CD8⁺ CAR T cells show enhanced infiltration and better tumor control in two different xenograft tumor models.**
**A–C** Experimental layout in (**A**). In (**B**), representative immunofluorescence images of anti-EGFR ctrl-CD8^CAR or Rapa-CD8^CAR cells (anti-CD3 Ab, green) distribution in tumor islets and stroma (tumor cells identified using a EpCAM Ab, red) in each microscope field (ctrl-CD8^CAR, $n = 17$ and Rapa-CD8^CAR, $n = 27$ microscope fields, MW Rank Sum Test). An ImageJ mask was used to identify tumor islets (EpCAM^pos), stroma (EpCAM^neg) and CAR T cells. Graphs on the right report the total amount of CAR T cells per section area (left) and the relative [tumor islets/stroma] CAR T cell ratio (corrected for the [tumor islets/stroma] area ratio). In (**C**), immunofluorescence images of lung tumor slices stained for EpCAM (red) and active-caspase-3 (green). The graph on the right shows the relative [active-caspase-3/EpCAM] ratio for each tumor islet (ctrl-CD8^CAR, $n = 136$ and Rapa-CD8^CAR, $n = 102$ tumor islets from different microscope fields, MW Rank Sum Test). **D** Experimental layout on top. Measurement of tumor size by bioluminescence (normalized to value at day of CAR T cell injection) in mice bearing lung tumors (derived from Luciferase^pos A549 cells i.v. injection 3 days before) untreated or infused with ctrl-CD8^CAR or Rapa-CD8^CAR cells (untreated, $n = 6$ mice; ctrl-CD8^CAR, $n = 8$ mice; Rapa-CD8^CAR, $n = 8$ mice, ANOVA on Ranks with posthoc Dunn's). **E** Experimental layout on top. Images and graphs were prepared as in B (ctrl-CD8^CAR, $n = 18$ and Rapa-CD8^CAR, $n = 14$ microscope fields, MW Rank Sum Test for graph on the left; unpaired two-tailed Student's T-Test for graph on the right) with the exception that an anti-CD45 Ab was used to identify CAR T cells. **F, G** Experimental layout on the left. Measurement of tumor size in A549-derived s.c. tumor-bearing mice untreated or infused with ctrl-CD8^CAR or Rapa-CD8^CAR cells (**F**; untreated, $n = 7$ mice; ctrl-CD8^CAR, $n = 8$ mice; Rapa-CD8^CAR, $n = 8$ mice, ANOVA on ranks with posthoc Dunn's). Amount of CD8⁺ CAR T cells per mg of tumor at endpoint (38 days) (**G**; ctrl-CD8^CAR, $n = 8$ and Rapa-CD8^CAR, $n = 7$ mice, unpaired two-sided Student's T-Test). Data are expressed as mean ± SEM in (**D**, **F**, **G**), and as box plot (center line, median; box limits, upper and lower quartiles; whiskers, 1.5x interquartile range; points, 5th and 95th percentiles) in (**B**, **C**, **E**). Scale bar, 100 μm in (**B**, **C**, **E**). Schemes in (**A**) and (**D–F**) have been created with BioRender.com.

since an excessive increase in mtROS amount would be detrimental for their migration because of the concomitant reduction in mito-ATP production. This is consistent with several studies showing that high ROS levels inhibit amoeboid-like motility[33]. Our data are also in line with previous reports suggesting that mito-ATP is required to support T cell motility in a 2D environment by providing energy for the acto-myosin contraction at the cell rear edge[13,16,58]. Low amounts of ROS, by modulating the redox status of many proteins, are positive contributors to T cell activation[59,60]. The exploration of the nature of the molecular targets of mtROS that control CD8⁺ T cell motility will be an important area for the future.

Several strategies modulating mitochondrial metabolism proved to be effective in increasing the efficacy of T and CAR T cell-based immunotherapy approaches[21,61–64]. The explanation to justify these effects relies on the ability of these treatments to increase T cell differentiation towards more memory/primitive-like states with superior long-term persistence[21,62,65,66]. However, the impact of these strategies on modulating T cell intratumoral motility or infiltration into tumor islets has never been addressed in these studies. Based on our findings, we propose that along with an improved memory T cell differentiation, the beneficial effects of these treatments can also be explained by an enhanced migration of T cells in tumors favoring the formation of productive contacts with cancer cells. In support of this, the combination of IL-7 and IL-15 cytokines, which are commonly used during CAR T cell production, promotes both memory T cell expansion and intratumoral T cell migration.

From a clinical perspective, we here propose to pharmacologically manipulate CD8⁺ CAR T cell metabolism during the in vitro expansion phase and prior to their injection into patients to confer on these cells a superior and long-lasting motility capacity while at the same time avoiding any possible alterations on tumor cell metabolism, which could unintentionally promote tumor growth. Alternative approaches to increase mitochondrial metabolism in CD8⁺ CAR T cells also consist of the genetic engineering of lymphocytes to overexpress metabolic enzymes[67]. However, we believe that the rapamycin-based strategy that we tested here would be preferable to genetic approaches permanently increasing mitochondrial activity, which would not allow T cells to engage different metabolic pathways when required. For example, during phases of cell proliferation, the OXPHOS rate must be reduced and ATP must be mainly produced by glycolysis to spare carbons for biosynthesis[27]. In this context, forcing T cells to permanently maintain a high OXPHOS rate could be detrimental. On the contrary, the rapamycin-based strategy did not alter basal respiration levels but only maximal respiratory capacity, indicating that CD8⁺ T cells maintain the flexibility to increase OXPHOS only when required (such as during migratory phases), while reducing it when needed (such as during proliferative phases). Unfortunately, given its role in inhibiting mTOR-dependent glycolysis, rapamycin treatment reduced the in vitro expansion rate of CD8⁺ CAR T cells. Although this may not be a limitation for most cancer patients, it is possible that this reduced proliferation rate would be an obstacle to generate enough CAR T cells from immunocompromised patients lacking a high initial quantity of T cells. In this case, alternative approaches not affecting proliferation could be preferable. For example, we tested bezafibrate treatment, which had no impact on cell proliferation, although its effects on 3D motility were less pronounced and no increase in mitochondrial mass was observed (therefore limiting its long-lasting efficacy). Another possibility would be to use nutrients capable of bypassing glycolysis to fuel OXPHOS (such as lactate, acetate, and pyruvate). This could be of particular interest in tumors where glucose uptake by T cells is severely inhibited. In sum, further studies will be required to implement the best metabolic reprogramming strategies enabling T cells to migrate actively, expand and survive within the tumor microenvironment.

Hypoxia is a major obstacle to anti-cancer approaches. Besides supporting cancer cell metabolic reprogramming and the Warburg effect[68,69], hypoxia also prevents T cells from mounting a potent anti-tumor response[70]. In this context, our data suggesting the prominent role of mitochondrial metabolism in supporting CD8⁺ T cell intratumoral motility are in line with the observation of a strong reduction in T cell motility within solid tumor hypoxic areas[71]. Therefore, it is possible that strategies aimed at increasing mitochondrial activity to support T cell intratumoral motility would not produce the expected results in a highly hypoxic TME devoid of $O_2$. In this case, alternative strategies should be envisaged. For example, culturing cells in mild acidosis increases ETC efficiency, allowing efficient OXPHOS even at low oxygen tension[72]. It would be interesting to test this strategy in a context of CAR T cells.

Metabolic alterations within the TME are known to impact T cell activities[73]. Our data suggest that long-time glucose or glutamine deprivation in the TME may negatively influence the ability of endogenous or CAR CD8⁺ T cells to properly infiltrate tumor islets. Surprisingly, FAs do not appear to be involved in supporting this process, at least when provided as directly available energy sources to migrating CD8⁺ T cells. Although metabolic strategies to sustain or rescue FAO proved to be effective in increasing TILs functions[46,64], these are mainly long-term treatments, presumably reprogramming metabolism to reinforce mitochondrial activity. In this regard, it would be interesting to test whether long-term FAs supplementation could have any effect on CD8⁺ T cell 3D motility. In addition, our data indicate that lactate can be used by CD8⁺ T cells to sustain intratumoral motility. Therefore, although lactate has been reported to play an inhibitory role for CD8⁺ T cells in the TME, especially for proliferation and cytotoxicity[74,75], our data suggest that its role may be more multifaceted, at least supporting motility in the absence of glucose. Of note, regulatory T cells (Tregs) can use lactate in the TME

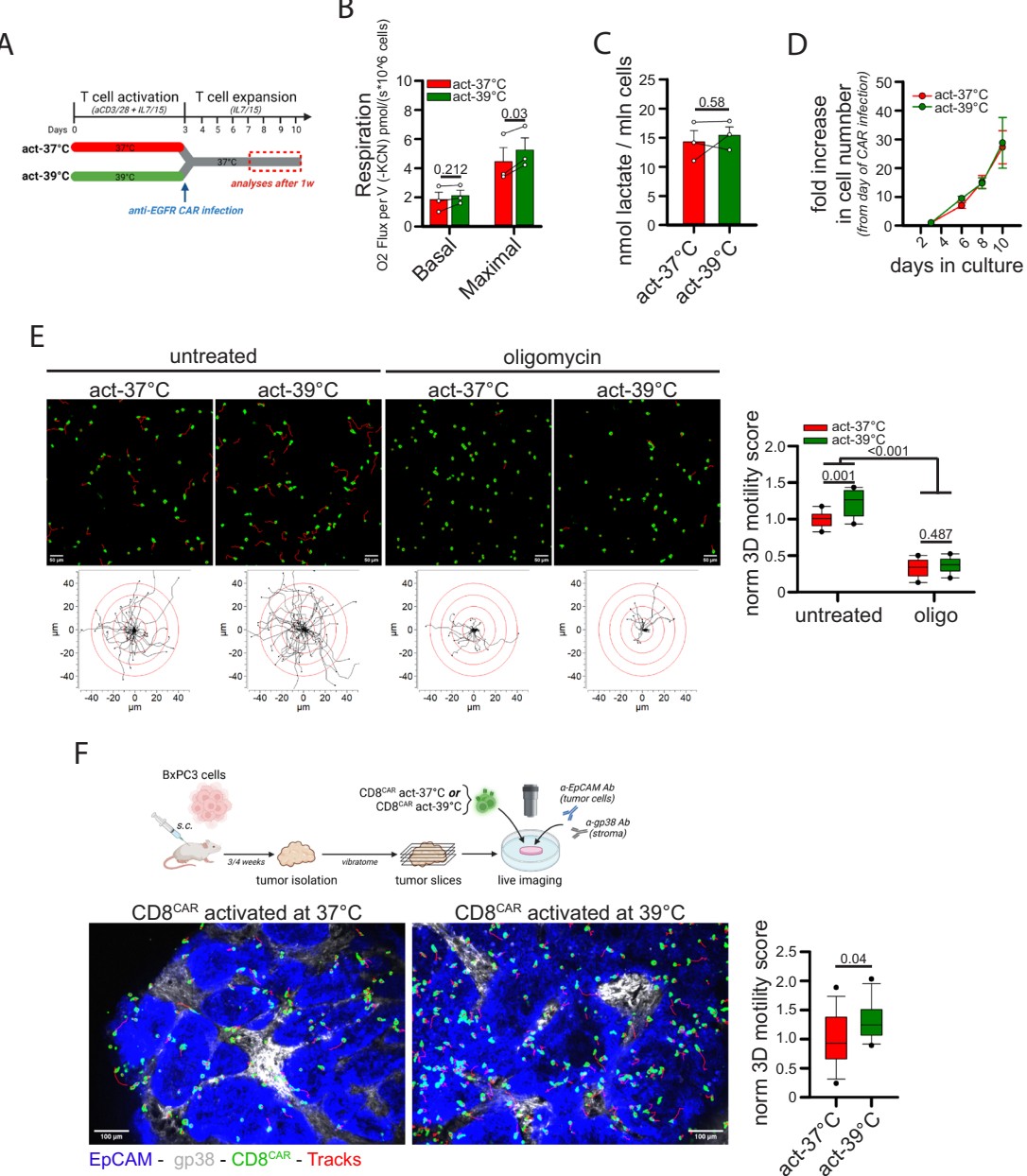

**Fig. 8 | CD8+ CAR T cells activated at 39 °C show higher mitochondrial respiration and better 3D intra-tumoral motility. A–E** Experimental layout in A. CD8+ T cells activated at 37 °C (act-37 °C) or at 39 °C (act-39 °C) for 3d have been placed back at 37 °C, infected with lentiviral particles to generate anti-EGFR CD8+ CAR T cells, and subsequently expanded at 37 °C. After 1 week, the following parameters were measured: basal and maximal respiration (**B**, $n = 3$ independent biological samples, unpaired two-tailed Student's T-Test), lactate production (**C**, $n = 3$, independent biological samples, unpaired two-tailed Student's T-Test), fold increase in cell number (**D**, $n = 3$ independent biological samples, Two-Way ANOVA), and normalized 3D motility Score (**E**, $n = 9$ movies, Two-Way ANOVA with posthoc Holm Sidak's). **F** Experimental layout on top. 3D motility of Calcein Green-stained act-37 °C or act-39 °C cells (cells in green, tracks in red) overlaid onto viable tumor slices derived from the BxPC3/NSG model (tumor islets in blue, EpCAM; stroma in gray, gp38) ($n = 16$ slices, unpaired Student's T-Test). Data are expressed as mean ± SEM in (**B**–**D**), and as box plot (center line, median; box limits, upper and lower quartiles; whiskers, 1.5x interquartile range; points, 5th and 95th percentiles) in (**E**, **F**). Scale bar, 50 μm in (**E**) and 100 μm in (**F**). Schemes in (**A**, **F**) have been created with BioRender.com.

to sustain their metabolism, providing them some advantage over conventional T cells (Tconvs)[76]. No study to date has addressed the relative distribution of Tregs and Tconvs within solid TMEs. Based on our results, it would be interesting to investigate whether lactate might enable Tregs to better infiltrate tumor masses compared to Tconvs cells (especially in a glucose-deprived TME) as a mechanism to support tumor growth.

In conclusion, we identified here the metabolic requirements of CD8+ T cells migrating in 3D environments, such as within a solid tumor mass. Our study provides a rationale for manipulating metabolism of CD8+ CAR T cells to improve their infiltration into tumor islets and potentiate their anti-tumor effect.

## Methods

### Study approvals

Human studies were carried out according to French law on biomedical research and to principles outlined in the 1975 Helsinki Declaration and its modification. Peripheral blood samples from adult individual were obtained from EFS (Etablissement Français du Sang) under agreement 18/EFS/030, including informed consent for research

purposes. Blood samples were anonymous and not associated with any personal data (including age, gender, or sex). Human lung biopsies from NSCLC adult patients were obtained from the Pathology Service of Cochin Hospital without any personal data (including age, gender or sex). Ethical procedures were approved by CPP Ile de France II (approval number 00001072, August 27, 2012) and INSERM Institutional Review Board (CEEI, IRB00003888, approval number 22/893, March 8, 2022), including informed consent for research purposes. NOD.Cg-Prkdc(scid)Il2rg(tm1Wjll)/SzJ (abbreviated NSG) immunodeficient mice were purchased from The Jackson Laboratory (Bar Harbor, ME, USA, 005557). Mice were housed in pathogen-free condition at Cochin Institute Animal Facility. Mice were kept in cages of no more than 5-6 mice each, divided by sex, under 12 h/12 h light/dark cycles, with standard temperature, humidity, and pressure conditions according to FELASA guidelines. Small squared mice house and paper were used for cage enrichment. Mice health was monitored by veterinary staff and health analysis for pathogens were performed according to FELASA guidelines. All efforts were made to minimize animal suffering and to reduce the number of mice used, in accordance with the European Communities Council Directive of 24 November 1986 (86/609/EEC). Mice were used for the experiments independently of sex and were sacrificed by neck dislocation. The mice protocol has been approved by Paris Descartes University (CEEA 17-039) and the French Ministry of Research (APAFiS #15076).

## Cell culture and reagents
Human peripheral blood mononuclear cells (PBMCs) were isolated using Ficoll gradient (800 g for 20 min, RT) and T cells were purified using Human Naïve CD8$^+$ T Cell Isolation Kit (Miltenyi, 130-093-244), Naïve CD4$^+$ T cell Isolation Kit (Miltenyi, 130-094-131) or Pan T cell Isolation Kit (Miltenyi, 130-096-535). Then T cells have been collected, washed, and used for subsequent analyses. Human T cells have been cultured in RPMI 1640 medium (Thermo Fisher 21875) supplemented with 10% Fetal Bovine Serum (Thermo Fisher A3840002), 2mM L-glutamine (Thermo Fisher 25030081) and 100 U/ml penicillin/streptomycin (Thermo Fisher 15140130). When indicated, glucose-(Thermo Fisher 11879) or glutamine-free (Thermo Fisher 31870) RPMI media were used. For in vitro activation, $8 \times 10^5$ naïve CD8$^+$ or CD4$^+$ T cells have been stimulated with 10 µg/ml anti-CD3 (plate-coated) (Biolegend 300465) and 2 µg/ml anti-CD28 (Biolegend 302943) for 72 h in 96well plate in presence of 25 ng/ml IL-7 and IL-15 (Miltenyi, 130-095-363 and 130-095-765), or in the presence of 200 U/ml IL-2 alone (Miltenyi, 130-097-746). Then, cells have been washed and expanded with ILs only. When indicated, cells were cultured in the presence of 10 nM rapamycin (Tocris 1292), 10 µM bezafibrate (Sigma Aldrich B7273), 1 mM AICAR (Tocris 2840) or 1 mM βNAD (Sigma Aldrich N1511). Drugs were removed on the day of assays and cells were washed at least twice with RPMI complete medium. Unless specified, these drugs were not present during the assays. To evaluate T cell motility in collagen gels or viable tumor slices, T cells were resuspended in a "*motility medium*" composed of RPMI 1640 medium supplemented with 2 mM glutamine, 0.5% Bovine Serum Albumin (BSA) (Sigma Aldrich A3059) and 10 mM HEPES (Thermo 15630). When specified, a glucose- or glutamine-free RPMI medium was used to prepare the *motility medium*. The following reagents were added either during motility assay or in cell culture for the indicated period of time before the assay: 11 mM glucose (Thermo Fisher A2494001), 2 mM or 11 mM glutamine, 85 µg/L Oleic Acid (BSA-conjugated, Aigma Aldrich O3008), 130–13000 µg/L Palmitate (BSA-conjugated, CaymanChemical 29588), 85-8500 µg/L Linoleic Acid (BSA-conjugated, Sigma Aldrich L9530), 5 µM etomoxir (Sigma Aldrich 236020), 10 µM trimetazidine (Sigma Aldrich 653322), 1 µM perhexiline (Sigma Aldrich, SML0120), 20 mM 2-deoxy-glucose (2DG) (Sigma Aldrich D8375), 100 µM 6,8-bis(benzylthio)octanoic acid (indicated as 6,8bOA, Sigma Aldrich SML0404), 10 µM CB-839 (GLSi inhibitor, Euromedex TA-T6797), 5 µM Compound

968 (GLS2 inhibitor, Sigma 352010), 10 µM AZD3965 (MCT1 inhibitor, MedChem Express HY-12750), 11 mM galactose (Sigma Aldrich PHR1206), 10 mM sodium lactate (Sigma Aldrich L7022), 1 mM sodium acetate (Sigma Aldrich 71188), 100 µM sodium pyruvate (Thermo Fisher 11360070), oligomycin (several doses, Sigma Aldrich 495455), rotenone (several doses, Calbiochem B48409), antimycin-A (several doses, Sigma Aldrich A8674), 3-nitropropionic acid (3-NP, several doses, Sigma Aldrich N5636), Atpenin-A5 (several doses, MedChemExpress HY-126653), MitoTEMPO (several doses, Sigma Aldrich SML0737), 500µM N-acetyl-cysteine (NAC, Sigma Aldrich, A9165), 1 µM PS-10 (Sigma Aldrich SML2418), 1 µM LDH inhibitor GSK2837808A (LDHi, Tocris 5189), 10 µM α-Lipoic Acid (Sigma Aldrich 62320), 1 mM succinyl phosphonate (MedChemExpress HY-12688A), 100 µM di-methyl-succinate (Sigma Adrich 112755), 100 µM di-methyl-α-ketoglutarate (Sigma Aldrich 349631), 500 nM CCCP (Alfa Aesar L06932.ME), and 50 nM ethidium bromide (Sigma Aldrich E1385). Human BxPC3 tumor cells (in-house stock) have been cultured in the same medium as T cells. Human A549 (mCherry$^{pos}$ and Luciferase$^{pos}$) and HEK-293T tumor cells (in-house stocks) have been cultured in DMEM medium (Thermo Fisher 11995) supplemented like RPMI medium for T cells. HUVEC cells (Thermo Fisher C0035C) have been cultured in Human Large Vessel Endothelial Cell Basal medium (Thermo Fisher M200500) supplemented with 10% FBS and Large Vessel Endothelial Supplement (Thermo Fisher A1460801). Cells were routinely tested for mycoplasma contamination.

## In vitro motility assay with collagen gel
T cells were washed once with DPBS, stained with 100 nM of Calcein Green (Invitrogen C34852) or Calcein Red Orange (Invitrogen C34851) for 20 min at 37 °C in DPBS, washed once in motility medium, resuspended in 50 µl at a final concentration of $20 \times 10^6$cells/ml and then mixed with 100 µl of collagen solution (PureCol Bovine Collagen, CellSystem 5005, previously neutralized using DPBS 10x) to reach a final collagen concentration of 2 mg/ml. Then, the mixture was loaded into an ibiTreat µ-Slide VI 0.4 slide (Ibidi 80606), which was placed at 37 °C in the incubator for 1 h to allow collagen polymerization. When performing nutrient depletion experiments, in order assure that the nutrients were correctly removed before the assay, T cells were first washed twice (each time for 5 min) with PBS. Then, cells were incubated 20 min in PBS containing the staining agent at 37 °C. After this incubation and in order to stop the reaction, we added in the tubes the motility medium which already contains the nutrients at the final assay concentration (*e.g.*, medium with glucose but without glutamine for experiments in Fig. 2G–I). Cells were washed further twice (each time for 5 min) with the motility medium. After a final centrifugation, cells were resuspended in the motility medium before adding the collagen. Then, the solution was added into Ibidi chambers and placed at 37 °C for collagen polymerization (approx. 1 h). Motility was always evaluated in the absence of any interleukin. Drugs or specific nutrients were added to the medium immediately before mixing cells and collagen. When indicated, telo-collagen (TeloCol purified Bovine Collagen, Cellsystem 5026) was used at a final concentration of 3 mg/ml. Live recording of T cell motility was performed using a Nikon epifluorescence microscope equipped with 37 °C heated-chamber, ORCA-Flash 4.0 camera (C11440 Hamamatsu) and Metamorph software (version 7.10.2.240, Molecular Devices). For each condition, 15 images (30 s time interval, 4/5 z-planes, z-distance 15 µm) were acquired using a 20x objective (binning rate, 2) at 3 different stages. In most cases, T cells from two different donors were stained with Calcein Green and Calcein Red and mixed before adding them to collagen. In other experiments, cells from different experimental conditions but from the same donor were stained with Calcein Green and Calcein Red and then mixed before adding collagen. Motility under hypoxia was performed using a spinning-disk (Yokogawa CSU-X1) Leica DM6000FS microscope equipped with a 37 °C heated-chamber, an ORCA-Flash 4.0

camera (C11440 Hamamatsu) and Metamorph software (version 7.8.9, Molecular Devices). Hypoxia was obtained by delivering 100% $N_2$ through a perfusion system into the culture medium in which the collagen gel was placed. An oximeter was used to estimate $O_2$ concentration, which was around 5% close to the collagen gel.

### Live imaging slice assay

Viable tumor slices were obtained from human NSCLC biopsies, BxPC3- or A549-derived tumors implanted subcutaneously into NSG mice or lungs of NSG mice inoculated i.v. with A549 cells (orthotopic lung tumor model). Slices were obtained and prepared essentially as previously described[77]. Briefly, tumor samples were cut into 1–2 $mm^3$ pieces, embedded into 7% agarose (Sigma Aldrich A0701) and cut into through vibratome (VT 1000 S, Leica) to obtain 300 μm-thick viable tumor slices (Leica Biosystem). Then, slices were stained with 200 ng/ slices of anti-EpCAM (BV421 or PE, clone 9C4, Biolegend 324220 or 324206) and anti-gp38 (eFluor660, clone 8.1.1, eBioscience 50-5381-82 or BV421, Biolegend 127423) Abs using Millicell 0.4 μm organotypic insert (Millipore PICMORG50) and metal rings for 30 min at 37 °C. NSCLC biopsies were also stained with 200 ng/slices of PE-anti-CD8 Ab (Biolegend 344706, clone SK1) to detect endogenous CD8+ T cells and Alexa647-anti-fibronectin Ab (BD Bioscience 563098, clone 10/Fibronectin) to detect stroma. When exogenous T cells were added to the slice, T cells stained with Calcein Green and/or Calcein Red (see above) were either (*i*) added onto the stained slice ($5 \times 10^5$ cells/30 μl of motility medium) for 20 min at 37 °C or (*ii*) injected into the tumor piece with a 30 G syringe ($10 \times 10^6$ cells/100 μl of motility medium) before embedding the tumor biopsy into agarose. Finally, slices were placed into a 30 mm petri dish (Thermo Fisher 121 V) filled with 5 ml of motility medium and let warm at 37 °C for 20 min in the incubator. Live recording of cell motility was performed using a spinning-disk (Yokogawa CSU-X1) Leica DM6000FS microscope equipped with 37 °C heated-chamber, ORCA-Flash 4.0 camera (C11440 Hamamatsu) and Metamorph software (version 7.8.9, Molecular Devices). For each condition, 15/20 images (5/10 z-planes, z-distance 15 μm) were acquired using a 10x or 25x objective (binning rate, 2) with a time interval of 30 s.

### Analysis of T cell motility

T cell motility was analyzed using ImageJ plugin TrackMate[78] (v.7.11.1) in a 3D environment (z-stack distance, 15 μm). LoG detector was used to identify spots (quality and maximal intensity filters applied for collagen assays) and Simple LAP Tracker to identify tracks (linking max distance 20 μm, closing-gap max distance 20 μm, frame gap 3). Only tracks containing 10 or more spots were considered for analysis. Motility Score was calculated as follows: (% of cells moving more than 10 μm as Euclidean distance)*(mean speed [μm/min] of moving cells). Images and movies shown throughout the manuscript are 2D z-stack reconstruction while analyses were performed on original 3D images (*i.e.*, also taking into account the z-axis displacement).

### Immunofluorescence on cells

For myosin staining, T cells migrating in collagen gel were fixed with 4% paraformaldehyde for 20 min at RT and then permeabilized with 0.5% Triton-X 100 (Sigma Aldrich T9284) for 15 min. For actin and tubulin staining, cells migrating in collagen gel were washed three times with PHEM solution (60 mM PIPES, 25 mM HEPES, 10 mM EGTA, 2 mM acetate) and then fixed with 4% paraformaldehyde + 0.2% glutaraldehyde + 0.1% Triton-X-100 in PHEM buffer for 15 min at RT. Cells were then incubated in blocking solution (1% BSA in DPBS) for 1 h. Primary antibodies (anti phospho-Ser-19 myosin Light chain 2, Cell Signaling 3671; or anti-tubulin, Sigma T9026, clone DM1A) were added o.n. in blocking solution at 4 °C (1:50 dilution). The following day, samples were incubated with secondary antibody (Alexa-488 anti-rabbit IgG, Invitrogen A32731; or Alexa568 anti-mouse IgG, Invitrogen

A10037) in presence of phalloidin-Alexa647 (Invitrogen A22287, to stain actin) for 1 h at RT in blocking solution. Finally, samples were washed, stained with DAPI, and mounted using Vectashield Mounting Medium (VectorLabs H-1900-10). Images were acquired using a spinning-disk (Yokogawa CSU-W1 T1) Ixplore IX83 microscope (Olympus) equipped with ORCA-Flash 4.0 V3 (sCMOS Hamamatsu) camera and CellSens Dimension software (Olympus, v4.1.1).

### Immunofluorescence on fixed samples

Samples were fixed o.n. with PLP solution (1% paraformaldehyde, 2 mM lysine-HCl, 550 mg/L $NaIO_4$) at 4 °C, cut into 1–2 $mm^3$ pieces, embedded into 7% agarose (Sigma Aldrich A0701) and then cut into 150 μm-thick sections through vibratome (VT 1000 S, Leica). For extracellular staining, slices were incubated 1 h at RT with 0.5% BSA in DPBS. For intracellular staining, slices were incubated for 20 min at −20 °C with 100% cold acetone (Carlo Erba 400971), washed twice with DPBS, and incubated for 1 h at RT with 1% BSA + 0.3% TritonX-100 (Sigma Aldrich T9284) in DPBS. Then, slices were placed onto Millicell 0.4 μm organotypic insert (Millipore PICMORG50) and 30 μl of 0.5% BSA / DPBS solution (for extracellular staining) or 0.5% BSA + 0.1% Tween20 (Sigma Aldrich P7949) / DPBS solution (intracellular staining) containing the desired Abs were added. A metal ring was used to keep the solution in place on the slice. Abs were incubated o.n. at 4 °C. The day after, slices were washed twice in DPBS and then mounted using Superfrost microscope slides (Epredia J1800AMNZ) and Vectashield antifade mounting medium (VectorLabs H-1900). The following antibodies were used (200 ng/slice): Alexa555-anti-TOMM20 (Abcam ab221292, clone EPR15581-54), Alexa647-anti-ATP5a (Abcam ab196198, clone EPR13030(B)), PE-anti-IDH2 (Abcam ab212122, clone EPR7577), Alexa488-anti-HK1 (Abcam ab184818, clone EPR10134(B)), Alexa488-anti-PFK2/PFKFB3 (Abcam ab203984, clone EPR12594), Alexa405-anti-GLUT1 (Abcam ab210438, clone EPR3915), DyLight405-anti-G6PD (Bio-Techne NBP2-22125V, clone 2H7), BV605-anti-EpCAM (Biolegend 324224, clone 9C4), PerCP-Cy5.5-anti-CD8 (Biolegend 344710, clone SK1), Alexa647-anti-CD8 (Biolegend 344726, clone SK1), FITC-anti-CD3 (Biolegend 344804, clone SK7), PE-anti-EpCAM (Biolegend 324206, clone 9C4), rabbit anti-cleaved caspase-3 (Cell Signaling 9664 S, clone 5A1E) and Alexa647-anti-rabbit secondary antibody (Thermo Fisher A-21245). Images were acquired using a spinning-disk (Yokogawa CSU-W1 T1) Ixplore IX83 microscope (Olympus) equipped with ORCA-Flash 4.0 V3 (sCMOS Hamamatsu) camera and CellSens Dimension software (Olympus, v4.1.1). To analyze protein expression in NSCLC samples, an ImageJ threshold mask was used to identify EpCAM+ areas (tumor islets) and CD8+ T cells. Then, "subtract" or "multiply" ImageJ commands were used to define CD8+ T cells in tumor stroma or islets, respectively. Protein expression (MFI) was calculated in the selected T cell populations.

### CRISPR-Cas9 gene editing

CRISPR/Cas9-mediated gene editing was performed on CD8+ T cells activated for 24 h. Briefly, 25 pmol of each specific crRNA was mixed with an equal amount of tracRNA (IDT technologies 1072534) and warmed at 95 °C for 5 min. Then, the solution was left cool down to RT and 15 pmol of Cas9 Nuclease V3 (IDT Technologies 1081059) were added. After 20 min incubation at RT, 16.4 μl P3 and 3.6 μl Supplement Reaction solutions from P3 Primary Cell 4D X Kit S (Lonza V4XP-3032) were added. Then, $5 \times 10^5$ T cells were washed once in DPBS and resuspended in 20 μl of the reaction mixture. Electroporation was performed using Amaxa 4D Nucleofector (Lonza), following the manufacturer instruction. After 30 min recovering at 37 °C, cells were restimulated for an additional 48 h. Target sequences (from IDT Technologies) used for gene editing were the following (three sequences were used together to target *PDHA1* and/ or *OGDH*): 5'-CACGGCTTTACTTTCACCCGGGG-3' (Hs.Cas9.PDHA1.1.AA), 5'-TATGCCAAGAACTTCTACGGGGG-3' (Hs.Cas9.PDHA1.1.AB), 5'-TAGAG-CAATCCCAGCGCCCAGGG-3' (Hs.Cas9.PDHA1.1.AC) 5'-TCCACGAGCTTG TCCACGTTGGG-3' (Hs.Cas9.OGDH.1.AA) 5'-TGGGACTAGTTCGAACTAT

GTGG-3′ (Hs.Cas9.OGDH.1.AB) 5′-ACGAATGCCGGAGCCCCACCGGG-3′ (Hs.Cas9.OGDH.1.AC).

## qRT-PCR for mitochondrial DNA

DNA extraction was performed with FastPure Blood/Cell/Tissue/Bacteria DNA Isolation Mini Kit (Vazyme Biotech DC112-01). Cells were lysed in the presence of proteinase-K (Sigma Aldrich 31158) at 56 °C, then DNA was isolated by silica gel columns purification technology. Quantitative real-time PCR was performed with LightCycler 480 SYBR Green I Master detection kit (Roche 04707516001) on a Light Cycler 480 machine (Roche). Three technical replicates were performed for each sample. Fluorescence curve analysis was performed using LightCycler 480 Software (version 1.5.1.62, Roche). Primer sequences were the following: ND1-Fw: 5′-CCCTACTTCTAACCTCCCTGTTCTTAT-3′ ND1-Rv: 5′-CATAGGAGGTGTATGAGTTGGTCGTA-3′ ND5-Fw: 5′-ATTTTATTTCTCCAACATACTCGGATT-3′ ND5-Rv: 5′-GGGCAGGTTTTGGCTCGTA-3′ 18S-Fw: 5′-AGTCGGAGGTTCGAAGACGAT-3′ 18S-Rv: 5′-GCGGGTCATGGGAATAACG-3′.

## Lactate assay

$0.5-2 \times 10^6$ cells have been washed and resuspended in 500 µl of *motility medium* in 96well plate for 4 h. Then, cells have been collected and lactate has been measured by using Lactate Assay Kit (Sigma MAK064), following manufacturer instructions. Measurements were performed using CLARIOstar instrument (BMG Labtech, software version 5.6.1).

## High-resolution respirometry with Oroboros device

The oxygen consumption rate (OCR) was measured in the *motility medium* (see above) by a high-resolution 2 mL glass chamber Oxygraph-2k respirometer (Oroboros). The electrode was calibrated at 37 °C, 100% and 0% oxygen before adding CD8⁺ T cells (2 mL at $2.5 \times 10^6$ cells/mL) to each chamber and the flux of $O_2$ consumption was measured in pmol/s. Oligomycin (0.5 µg/mL) was added to block complex V, as an estimate of the contribution of mitochondrial leakage to overall cellular respiration. Increasing amounts of carbonyl cyanide m-chlorophenyl hydrazone (CCCP) were added to measure maximum respiratory capacity. Finally, potassium cyanide (KCN) was added to measure the non-mitochondrial respiration.

## Measurement of $CO_2$ flux

$15 \times 10^6$ CD8⁺ T cells were incubated in RPMI 1640 medium (Thermo Fisher 31870) containing 11 mM [U-¹⁴C]-glucose (0.1 mCi/mmol) or in RPMI 1640 medium (Thermo Fisher 11879) containing 2 mM [U-¹⁴C]-glutamine (0.1 mCi/mmol). After 6 h, the media were transferred to a conical glass vial for $CO_2$ production measurement. Perchloric acid was injected into the incubation media through the rubber cap to a final concentration of 4% (v/v). Benzethonium hydroxide was injected through the rubber cap into a plastic well suspended above the incubation media. During 1 hour of vigorous shaking at 25 °C, the released [¹⁴C]-$CO_2$ was trapped by the benzethonium hydroxide and assessed by scintillation counting.

## Flow cytometry

The following antibodies have been used to stain extracellular proteins (1:100 dilution): BV711 anti-CD45RA (Biolegend 304138, clone HI100), APC anti-CD45RO (Biolegend 304210, clone UCHL1), PE anti-CCR7 (Biolegend 353204, clone G043H7), APC-Cy7 anti-CD27 (Biolegend 302816, clone O323), PE-Cy7 anti-CD95 (Biolegend 305621, clone DX2), CD49D-PE (Biolegend 304303, clone 9F10), CD62L-PerCP (Biolegend 304824, clone DREG-56), CD2-APC (Miltenyi 130-116-253, clone REA972), CCR5-A488 (Biolegend 359103, clone J418F1), CXCR3-PE (Biolegend 353706, clone G025H7), CXCR4-PerCP (Biolegend 306515, clone 12G5), CD44-A647 (Biolegend 397512, clone C44-Mab5), CD103-APC (Biolegend 350216, clone Ber-ACT8), cd11a-PE/Cy5 (Biolegend

301210, clone HI111), CCR3-PE (Biolegend 310705, clone 5E8), LPAM-1-APC (eBioscience 17-5887-82, clone DATK-32), GLUT1-PE (R&D System FAB1418P, clone 202915). True-Nuclear Transcription Factor Buffer Set (Biolegend 424401) was used to stain intracellular proteins, detected with the following anti-human antibodies (1:50 dilution): PE-anti-GrzB (Biolegend 372208, clone QA16A02), PE-Cy7-anti-Perforin (Biolegend 353315, clone B-D48), APC-anti-IFNγ (Biolegend 502511, clone 4 S.B3), BV421-anti-TNF (Biolegend 502932, clone Mab11), Alexa647-anti-ATP5a (Abcam ab196198, clone EPR13030(B)), PE-anti-IDH2 (Abcam ab212122, clone EPR7577), Alexa488-anti-HK1 (Abcam ab184818, clone EPR10134(B)), Alexa488-anti-PFK2/PFKFB3 (Abcam ab203984, clone EPR12594), Alexa405-anti-GLUT1 (Abcam ab210438, clone EPR3915), DyLight405-anti-G6PD (BioTechne NBP2-22125V, clone 2H7), anti-MCT1 (BioTechne FAB8275T, clone 882616). Primary antibodies were incubated o.n. at 4 °C. To evaluate GranzymeB, Perforin, IFNγ and TNF production, $1 \times 10^6$ cells have been restimulated for 5 h in a 96well plate with 3 µl of TransAct (Miltenyi 130-111-160). 5 µg/ml BrefeldinA (Biolegend 420601) has been added for the last 2 h and a half. For T cell subpopulations, the following subsets were defined by flow cytometry gating strategy: stem cell memory-like (Tscm, CD45RO^neg CD45RA^pos CD27^pos CCR7^pos CD95^pos); naïve-like (naive, CD45RO^neg CD45RA^pos CD27^pos CCR7^pos CD95^neg); effector-like (Teff, CD45RO^neg CD45RA^pos CD27^neg CCR7^neg); central memory-like (Tcm, CD45RO^pos CD45RA^neg CD27^pos CCR7^pos); effector memory-like (Tem, CD45RO^pos CD45RA^neg CD27^neg CCR7^neg). Cells outside the indicated gates were defined as "others". Gating strategy is reported as supplementary file (Supplementary Fig. 13). To evaluate glucose uptake, mitochondrial membrane potential (MMP), mitochondrial mass or mitochondrial ROS (mtROS), cells were incubated for 20/30 min at 37 °C in the presence of 30 µM 2-NBDG (Thermo N13195), 100 nM TMRE (Thermo Fisher T669), 100 nM MitoTrackerGreen (Thermo Fisher M7514) or 2.5 µM MitoSox (Thermo Fisher 11579096), respectively. Then, cells have been washed and analysed. Mitochondrial mass (MitoMass) has been calculated as (MitoTrackerGreen/FSC) ratio. MMP has been calculated as (TMRE/MitoTrackerGreen) ratio, mtROS amount as (MitoSox/FSC) ratio. Samples were analysed using BD Accuri C6 (CFlow Plus software v1.0.264.15) or BD Fortessa (FACSDive software v6.1.3).

## Infiltration into collagen gel

$5 \times 10^5$ human BxPC3 tumor cells were resuspended into 450 µl of complete RPMI medium plus neutralized collagen solution (2 mg/ml) and inserted into one side chamber of µ-slide Chemotaxis Chamber (Ibidi 80326). The other side chamber was filled with medium alone. After 24 h, T cells were stained with Calcein Green or Calcein Red as described above, mixed in a 1:1 ratio, and $2 \times 10^6$ cells were resuspended into 80 µl and inserted into the middle chamber of the µ-slide. Z-stack images were acquired at 0 h and 2 h. ImageJ software was used to identify cells (as ROIs) and to calculate the minimum distance from starting area.

## Anti-EGFR CAR and PercevalHR lentiviral infection

To generate anti-EGFR CAR, downstream of a signal peptide scFv sequences derived from nimotuzumab Ab (IMGT/2Dstructure-DB INN 8545H, 8545 L) were introduced into the expression cassette encoding CD8 hinge and transmembrane domains (194–248 Aa, GenBank: AAH25715.1), 4-1BB costimulatory domain (214-255 Aa, GenBank: AAX42660.1), CD3z signaling domain (52-163 Aa, GenBank: NP_000725.1) and GFP coding sequence (downstream of IRES sequence). Production of lentiviral particles was performed by transient co-transfection (polyethylenimine:DNA at 3.5:1 ratio) of HEK 293 T cells with the second-generation packaging system plasmid psPAX2 and pMD2.G (Addgene, 12260 and 12259). Viral supernatants were harvested at 48 h post-transfection. Viral particles were concentrated by ultracentrifugation at 25.000 g for 2 h and conserved at

−80 °C. Anti-EGFR CD8⁺ T cells (defined as CD8$^{CAR}$) were generated by infecting $6 \times 10^5$ CD8⁺ T cells activated for 72 h with 50 μL (MOI2) of lentiviral particles in 600 μL of RPMI complete medium (supplemented with IL-7 + IL-15) in 48well plate. After 3 days, cells have been collected, washed, and used for subsequent analysis. PercevalHR plasmid[39,40] (Addgene 49083) was used to generate lentiviral particles using the same procedure as for "CAR" particles. After 1 week of stimulation, cells were stained with Calcein Red, and motility was analysed in 2 mg/ml collagen gel. PercevalHR and Calcein Red signals was measured at timepoint 0 and then the ratio was calculated. Cells were defined as motile or immotile by visual inspection following motility analysis of the TrackMate software.

## Cytotoxicity assay

Anti-EGFR CAR T cell cytotoxicity has been evaluated using mCherry$^{pos}$ A549 cells. Briefly, A549 cells were plated at $1 \times 10^5$ cells/250 μl on μ-Slide 8 Well slides (Ibidi 80826) in DMEM medium. After 24 h (time 0 h), anti-EGFR CAR T cells were added to obtain a CAR:target ratio of 2.5:1, 5:1 or 10:1 in RPMI complete medium. A549 cell viability was evaluated by quantifying mCherry fluorescence (MFI) at different time points (24 h, 48 h) with a fluorescent microscope and normalized to MFI values in absence of CAR T cells.

## Orthotopic and subcutaneous tumor models

For the orthotopic tumor model, $2 \times 10^6$ human A549 cells (mCherry⁺ Luciferase⁺) were injected i.v. into the tail vein of 2 months-old NSG nude mice (independently of sex). After 3 weeks, mice were either (i) sacrificed to obtain lung slices to be used in motility experiments (see above) or (ii) infused i.v. with $5 \times 10^6$ anti-EGFR CD8⁺ CAR T cells into the tail vein. After 4 days, mice were sacrificed, and lungs were collected. For flow cytometry, lungs were digested with DNase (Roche 4536282001) and Liberase (Roche 5401119001) solution at 37 °C for 1 h and then smashed using a 70 μm Cell Strainer filter (Corning 352350). Collected cells were stained with Alexa647-anti-CD8 (Biolegend 344726) and PE-anti-EpCAM (Biolegend 324206) Abs for flow cytometry analysis (BD Accuri C6). For immunofluorescence, lungs were fixed in PLP solution (see above) for 24 h at 4 °C and, cut into 1-2mm³ pieces, embedded into 7% agarose (Sigma Aldrich A0701), and sliced using a vibratome (VT 1000 S, Leica) to obtain 150μm-thick sections. Slices were processed using FITC-anti-CD3 (Biolegend 344804) and PE-anti-EPCAM (Biolegend 324206) Abs. To follow tumor growth, 3 days after i.v. injection of A549 cells, mice were injected i.v. with $5 \times 10^5$ anti-EGFR CD8⁺ CAR T cells into the tail vein. Tumor growth in the lungs was followed by bioluminescence using PhotonIMAGER system (Biospacelab) and analysis was performed using M3Vision software (Biospacelab, software version 1.2.1.32016). To assess tumor burden in the orthotopic model, an evaluation grid was implemented according to our ethical committee in order to sacrifice mice reaching a defined endpoint threshold (calculated by taking into account tumor bioluminescence score, mouse behavior, physical appearance, weight, etc.). None of the mice reached the endpoint threshold score at the end of the experiment. For the subcutaneous tumor model, $5 \times 10^6$ human A549 cells were injected s.c. into the right flank of 2 months-old NSG nude mice (independently of sex). To evaluate CAR T cell infiltration, mice were injected i.v. after 5/6 weeks with $5 \times 10^6$ anti-EGFR CD8⁺ CAR T cells and sacrificed 4 days after to perform the same analyses described for orthotopic tumors. Alternatively, 10 days after A549 cell injection, A549-derived tumor-bearing mice were i.v. inoculated with $5 \times 10^6$ anti-EGFR CD8⁺ CAR T cells and tumor growth was followed through caliper measurement for additional 4 weeks. Then, mice were sacrificed, tumors were digested (like what described above for lungs), and collected cells were stained with PE-anti-EPCAM (Biolegend 324206) and Alexa488-anti-CD45 (Biolegend 304017) Abs for flow cytometry analysis (BD Accuri C6). We did not exceed the maximal s.c. tumor size permitted by our ethics committee (1500 mm³).

For the competition assay, anti-EGFR ctrl and Rapa CD8⁺ CAR T cells were stained with 1 μM CFSE (Invitrogen C34554) or 5 μM Cell Proliferation Dye eFluor670 (eBioscience 65-0840-90) and then mixed in a 1:1 ratio before being injected into tumor-bearing NSG mice (lung or s.c. tumor models as indicated before). After 3–4 days, the mice were sacrificed and infiltration into lung or s.c. tumors was assessed by flow cytometry (only for the s.c. model) or immunofluorescence on vibratome-cut slices using a spinning-disk (Yokogawa CSU-X1) Leica DM6000FS microscope equipped with a 37 °C heated-chamber, an ORCA-Flash 4.0 camera (C11440 Hamamatsu) and Metamorph software (version 7.8.9, Molecular Devices).

Subcutaneous tumors were also obtained through injection of human $5 \times 10^6$ BxPC3 cells s.c. into the right flank of 2 months-old NSG nude mice (independently of sex). After 3/4 weeks, mice were sacrificed, and tumors were collected to obtain viable tumor slices used in the slice assay (see above). We did not exceed the maximal s.c. tumor size permitted by our ethics committee (1500 mm³).

## Western blot

Western blots were performed as previously described in ref. 79. Briefly, cells have been lysed in RIPA buffer 20 min on ice (Tris-HCl 50 mM pH8, NaCl 150 mM, Nonidet-P40 1%, Sodium deoxycolate 0.5%, SDS 0.1%) supplemented with Protease Inhibitor Cocktail (Thermo Fisher 78429) and 1 mM NaVO₄. Samples were then centrifuged at 4 °C 20 min and denatured 5 min at 95 °C in denaturation buffer (with 5% beta-mercaptoethanol). Samples were run on 4–12% polyacrylamide gels (Invitrogen NW04125BOX) and transferred on PVDF membranes (Thermo Fisher 88518) using Mini-Trans Blot Cell (Bio-Rad 1703930). Membranes were blocked in 3% w/v non-fat dry milk containing 0.1% Tween20 (Sigma-Aldrich) and incubated with the appropriate antibodies o.n. at 4 °C. The following primary anti-human antibodies have been used (1:1000 dilution): anti-actin (Cell Signaling 4970, clone 13E5), anti-OGDH (Sigma Aldrich HPA019514), anti-Hsp90 (Cell Signaling 4877 S, clone C45G5), anti-PDHA1 + A2 (Cell Signaling 2784 S) and OxPhos Human WB Antibody Cocktail (Thermo Fisher 45-8199). Detection of protein signals was performed using ECL Prime Western Blotting Detection Reagent (Cytiva RPN2236) and Fusion FX instrument (Vilber Lourmat, software version 16.16.0.0). The original scans of the western blots presented in the manuscript are reported as supplementary file (Supplementary Fig. 14).

## Transendothelial migration assay

5μm-pore transwell inserts (Costar 3421) have been coated for 1 h at 37 °C with gelatin (Gibco S006100) and 10 μg/ml bovine fibronectin protein (Thermo Fisher 33010018). Subsequently, $5 \times 10^5$ HUVEC cells were added into the insert, let expanded for 24 h and stimulated o.n. with 10 ng/ml human TNF (Thermo Fisher PHC3016). The day after, CAR T cells were stained with either Calcein Green or Cell Proliferation Dye eFluor670 (Thermo Fisher 65-0840-85) (same procedure used for motility assay, see above) and mixed in a 1:1 ratio. Then, $2 \times 10^5$ cells were added to the transwell upper chamber in a motility medium (BSA, no FBS). Transwell lower chamber was filled with RPMI complete medium (10% FBS). The transmigrated cells were collected and quantified by BD Accuri C6 cytometer.

## Statistical analysis

In the Figure legends, "n" indicates either the number of biological independent samples, donors, the number of movies (no more than 3 from the same donor) or the number of mice used. Data are expressed as mean ± SEM or as box and whiskers plot. Comparisons between normal groups were done using paired or unpaired two-tailed Student's T-test (two groups) or One-way and Two-way ANOVA. Mann-Whitney Rank Sum Test (unpaired), Wilcoxon Signed Rank Test (paired) or ANOVA on Ranks were used if samples did not meet assumptions of normality. Adjustments for pairwise comparisons were

performed using post hoc tests. Statistical analysis has been performed using SigmaPlot (v15).

### Reporting summary

Further information on research design is available in the Nature Portfolio Reporting Summary linked to this article.

## Data availability

Raw immunofluorescence images of NSCLC samples generated in this study have been deposited on Zenodo with accession number [10.57889] and DOI number (https://doi.org/10.57889/Simula.et.al.2023) with unrestricted access. All other data supporting the findings of this work are available in the main text, the Supplementary Information, and Source Data files. Source data are provided with this paper.

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

## Acknowledgements

This work was partially supported by the European Union's Horizon 2020 research and innovation programme under the Marie-Skłodowska Curie grant agreement No. 945298 "ParisRegionFP" (LS), the French "Ligue Nationale contre le Cancer" (Equipes labellisées, ED), GeFluc Ile-De-France (Call 2021 & Call 2023, LS), European Union's Horizon Europe programme (HE-MSCA-PF 101062302, LS), and Fondation ARC (PDF2 ARCPDF22020010001123, LS). LS and MCAG were also supported by a grant from Institut Cochin (PIC 2022). MF and DA were respectively supported by "Ligue Nationale contre le Cancer" and "China Scholarship

Council" PhD fellowships. We are extremely thankful to members of the Cochin Institute Imaging (IMAG'IC), Cytometry and ImmunoBiology (CYBIO) and In vivo Imaging (PIV) facilities for their help and support in performing microscopy, flow cytometry and animal experiments, respectively.

## Author contributions

Conceptualization, Funding Acquisition & Writing - Original Draft: L.S. and E.D. Investigation: L.S., M.C.-A.G., M.F., D.A., F.P., G.B., N.B., A.R. and I.R. Formal Analysis: L.V. Resources: D.D., A.L.M., M.A., P.I., D.A. and F.P. Writing - Review & Editing: M.C.-A.G., F.P. and P.I. All authors approved the manuscript.

## Competing interests

The authors declare no competing interests.
