## [Peer Review File · Nature Communications]

Mitochondrial metabolism sustains CD8+ T cell migration for an efficient infiltration into solid tumorsREVIEWER COMMENTS

Reviewer #1 (Remarks to the Author):

Mitochondrial metabolism sustains CD8+ T cell migration for an efficient infiltration into solid tumors

The manuscript by Simula et al., identifies important metabolic sources fueling T cell migration in 3D models. The authors used various inhibitors and genetic approaches to nicely present data supporting that 3D migration was dependent on the TCA cycle, and T cell metabolism could be manipulated during ex vivo expansion to support enhanced migration to tumors in vivo. The study suggests that an improved understanding of metabolic pathways governing CD8 T cell motility in the TME could be used therapeutically to overcome dynamic TME metabolic alterations to improve CD8 T cell migration.

This is a well-performed, interesting paper addressing an important question in the field of T cell biology and immunology – with exciting therapeutic implications.

I don't think further major experiments are required for publication; addressing the following comments would strongly improve the scientific and translational interpretations in this paper.

1. In vitro 3D migration assay with collagen gel is a key experimental condition for the study; thus it is especially important that the basic characterization is not only sound but also complete in every aspect. It was not entirely clear whether there were any metabolic components present in the collagen gel. Any additional trace amount of nutrients in the system may impact these (seemingly metabolically sensitive) experimental conditions.
2. (Like it was addressed in Fig 3) many of the changes in vitro assays in Fig 1 & 2 may be due to changes in T cell populations (not due to T cell intrinsic changes). Therefore, it would be important to check the T cell heterogeneity pre vs post treatments (e.g. mitochondria phenotypes, GLUT expression, integrins & chemokine expression, etc).
3. The authors, while testing the effects of many key metabolic inhibitors and limited glucose and glutamine, never examined the effects of hypoxia vs normoxia. The

experiments strongly implicate a reliance on the TCA cycle and ATP production driven by glycolysis and glutaminolysis, which requires oxygen. All may be limited in most TME's. Although many in vitro conditions are slightly hypoxic, the authors could determine the effects of rapamycin treatment on motility in more controlled conditions of hypoxia, or perhaps by the hypoxia-mimetic CoCL₂. In most experiments, T cells were either overlaid onto tumor slices or recovered shortly after injection within tumors and then imaged presumably in normoxia. In this method, it is not possible to determine how microenvironmental factors, such as hypoxia, are influencing motility.

4. Additionally, the authors do not really address the T cell intrinsic factors such as smGTPases, MTOC and cytoskeletal rearrangements in the context of these treatments which would strengthen the (direct) connection between metabolism and migration compared to motility.

5. The authors show that rapamycin treatment of CAR T cells may increase CAR T cell infiltration into tumor islets in either s.c. or lung metastasis models. The authors however do not clearly show that the enhanced intratumoral motility is due to the increased mitochondrial metabolism of these cells. Is this infiltration directly due to the changes in metabolism, or is it something else, for instance rapamycin treatment changed the differentiation status of the T cell product. It may also influence the chemokine receptor profile. The authors should compare the expression of key homing markers on their control and rapamycin treated CAR T cells. Perhaps limiting mitochondria during expansion would have the opposite effect.

6. A direct comparison of ctl vs rapamycin treated cells via competition assays in vivo would strongly support the findings in this paper.

7. There is a difference in tumor islet infiltration compared to solid tumor infiltration and the language is not always clear. Would have been nice to see characterization of hypoxia and nutrient density in tumor islets.

Reviewer #2 (Remarks to the Author):

The manuscript submitted by Simula et al., sheds new light into our understanding of metabolic reprogramming and T cell motility. This work will appeal to the adoptive immunotherapy audience as the ability of T cells to infiltrate solid tumors and reach cancer

cells is a critical aspect of efficacy. In an important advance, the authors uncover the relative importance of glycolysis versus OX PHOS in proliferation versus motility, respectively. Distinct metabolic requirements of T cell responses were previously unclear as it was assumed that overall metabolic flexibility would support all needs. The authors include excellent models such as 3D collagen gels and tumor slices to demonstrate the preferential reliance on mitochondrial metabolism on motility within tumors. Remarkably, the paper reveals a novel role of lactate in T cell motility (in medium lacking glutamine and conditioned with 2DG). These observations should be extended to pyruvate, and acetate. As they can support mitochondrial metabolism when glucose is low, can they recover motility in 2DG-treated cells cultured in glutamine-free medium? Alternatively, it would be interesting to see if MCT1 inhibition reverses the beneficial impact of lactate.

In another advance, the authors develop a model in which the contingency energy (respiratory) reserve supports T cell motility in tumor slices. By impairing maximal respiratory function with OA, the migratory capacity is limited providing a novel link between the contingency energy reserve in the mitochondria and T cell migration. IL-7/15 which are increasingly used to condition T cells for cell-based therapies, potentiate mitochondrial function and this carries over to enhanced motility as expected. Here the authors show that T cells that can effectively infiltrate tumor islets demonstrate enhanced expression of IDH2, ATP5a and TOM20 exemplifying the importance of mitochondrial metabolism in T cells within tumors.

Overall the paper has the potential to be translationally relevant, and scientifically appealing but there are a number of weaknesses that need to be addressed.

Main Points

Given the mechanism of action of CB839, and its selectivity for GLS-1, the data shows that T cells with an intact GLS-2 isoform cannot compensate sufficiently to support motility. Can the authors discuss this in more detail? And can they show what happens to max oxygen consumption rates/mitochondrial function when CB839 is employed?

In figure 7D, the anti-tumor function of rapa- CAR T cells are analyzed in an in vivo xenograft

model (A549 IV infused). There are inherent limitations with this model as the standard CAR performs so poorly and the differences in tumor control of rapa-treated CAR T cells are not obvious. The numbers of untreated mice (n=2), CAR (n=3) and rapa CAR (n=3) are very low for an in vivo experiment. It's also unclear if this model was repeated more than once? The tumor model needs to be repeated.

In supplementary figure 10 (relating to Figure 7), looking at tumor growth in individual mice, 4/6 of the untreated mice have similar levels of tumor growth as their CAR counterparts. It's also unclear if this experiment was repeated more than once. A simple killing assay in vitro would be informative to determine if rapa-treated EGFR CAR T cells have enhanced tumor control. Additionally, a measurement of EGFR expression in A549 cells (target cell line) would be beneficial as overexpression maybe needed to boost CAR-mediated cytotoxicity.

It would be interesting to see if OA leads to a reduction in TCA cycle metabolites acetyl CoA, citrate, succinyl CoA.

The conditioning strategy at 39°C is intriguing. How does activation at 39°C impact subsequent CAR transduction? i.e. it more/less permissive for CAR lentiviral infection?

Minor Points

As described above, it's unclear from the figure legend how often the xenograft models were performed.

The authors refer to 6,8 bis(benzylthio) octanoic acid as octanoic acid in the materials section. I think this will cause confusion in the main text. The former is an inhibitor of PDH and OGDH; the latter is an 8-carbon medium chain FA species which fuels the TCA cycle.

An important point for future studies relates to the toxicity, and variability of CCCP for human T cell cultures. A shift to BAM15 as an uncoupling agent would be preferable.

Supplementary figure 1 (A) it looks like the largest fold increase occurred without glucose

and glutamine. Is this mislabeled?

In supplementary figure 10 (looking at the individual growth curves), 4/6 of the untreated mice have similar levels of tumor growth as their CAR counterparts.

In supplementary figure 10C, the legend refers to the xenograft model in Figure 7E. It's actually referring to Fig. 7F.

I think it is important to note that the scenario described in this paper (such as the mitochondrial adaptations observed in IL-7/15 treated T cells) and potentiated mitochondrial function, maybe context specific (lung cancers). Oxygen levels in the lung may permit/support mitochondrial function that in turn facilitates enhanced motility. In other models such as melanoma, and ovarian cancer, mitochondria function is severely suppressed, particularly in the CD8 T cell subset. As noted in their discussion, this paper will seed future efforts to address how motility can be supported in hypoxic TMEs such as these. Mindful to add, glucose uptake is also severely inhibited in TILs (B16 melanoma models). From this perspective, exploring how other fuels such as pyruvate, acetate, and lactate support motility are important future directions.

Reviewer #3 (Remarks to the Author):

This manuscript investigates the metabolic requirements for T cell migration in solid tumors. Overall, there is a massive amount of experiments and data in this paper. While the differences are not dramatic, many are significant, and multiple inhibitory approaches are taken, to support a link between mitochondrial metabolism and T cell migration in solid tumors. Ultimately, much of the data supports that enhancing ATP and mitochondria-based metabolism supports T cell migration in human tumor slices and xenograft tumor models. Two previously reported approaches (rapamycin or heat treatment of engineered T cells prior to adoptive cell therapy) known to promote mitochondrial fitness are tested here and are shown to enhance T cell migratory capacity in tumor nests and ultimately antitumor activity. Overall, the data is supportive of the hypothesis and the authors are fair in their interpretations. Some issues remain including model and system choice as well as novelty

overall.

Major Issues

- Many of the results provided are correlative and/or supportive but not clearly mechanistic in nature.
- No immunocompetent animal models are tested and thus the restricted use of xenografts is a limitation.
- Text requires major editing for conciseness and correct grammar, tenses, etc.
- The authors likely test every inhibitor available to target various metabolic pathways in T cells. Is there a way the authors can make the text and figures more linear and understandable to a broader audience. Is it necessary to show all the negative data in the main figures and restrict labels to acronyms of inhibitors? Or can the data be organized in a more logical manner with headings for specific pathways targeted.
- Assessment of migration of polyclonal, non-tumor reactive T cells in a tumor microenvironment is a limitation in the early part of this study and is somewhat a limitation. Unclear why non-specific T cells are being evaluated.
- Are the metabolic properties identified general principles for effector T cell migration regardless of the tumor microenvironment?
- Is it not already known that cellular migration requires ATP? This perhaps is not surprising.
- The approaches to enhance mitochondria metabolism in T cells are not particularly novel.
- While much of the results are significant, many graphs are either not significant or barely significant.

Minor Issues

- Authors may consider to temper the interpretations and replace words like “indicate” with “support”

Reviewer #1 (Remarks to the Author):

Mitochondrial metabolism sustains CD8+ T cell migration for an efficient infiltration into solid tumors
The manuscript by Simula et al., identifies important metabolic sources fueling T cell migration in 3D models. The authors used various inhibitors and genetic approaches to nicely present data supporting that 3D migration was dependent on the TCA cycle, and T cell metabolism could be manipulated during ex vivo expansion to support enhanced migration to tumors in vivo. The study suggests that an improved understanding of metabolic pathways governing CD8 T cell motility in the TME could be used therapeutically to overcome dynamic TME metabolic alterations to improve CD8 T cell migration.

This is a well-performed, interesting paper addressing an important question in the field of T cell biology and immunology – with exciting therapeutic implications. I don't think further major experiments are required for publication; addressing the following comments would strongly improve the scientific and translational interpretations in this paper.

We would like to thank the Reviewer for her/his general appreciation of our work.

1. In vitro 3D migration assay with collagen gel is a key experimental condition for the study; thus it is especially important that the basic characterization is not only sound but also complete in every aspect. It was not entirely clear whether there were any metabolic components present in the collagen gel. Any additional trace amount of nutrients in the system may impact these (seemingly metabolically sensitive) experimental conditions.

We fully agree with the Reviewer that the nutrient composition in the collagen gel is a parameter that needs to be precisely controlled during this assay. This is particularly true for the nutrient deprivation conditions, in which we tested the motility in the absence of one or more nutrients that were previously present during cell culture (Fig. 1B-C-D, Fig. 2A-G-H-I-J, Fig. S1A-B-C-G, Fig. S2D-E-F-G). We have modified the Material and Methods section (see lines 677-686) to precisely describe the protocol of the 3D migration assay in nutrient deprivation conditions. It reads as follows:

“When performing nutrient depletion experiments, in order assure that the nutrients were correctly removed before the assay, T cells were first washed twice (each time for 5min) with PBS. Then, cells were incubated 20min in PBS containing the staining agent at 37°C. After this incubation and in order to stop the reaction, we added in the tubes the motility medium which already contains the nutrients at the final assay concentration (e.g., medium with glucose but without glutamine for experiments in Figs. 2G-H-I). Cells were washed further twice (each time for 5min) with the motility medium. After a final centrifugation, cells were resuspended in the motility medium before adding the collagen. Then, the solution was added into Ibidi chambers and placed at 37°C for collagen polymerization (approx. 1h).”

Overall, the whole procedure takes 2 to 3h to be completed (including collagen polymerization) and during all this time before the assay the cells were placed first in PBS and then in the final medium composition (with only the correct nutrients) at least since 1h and half before recording their motility. We believe that this time window should be sufficient to let the cells adapt to the new nutrient composition and to assure that no traces of undesired nutrients are present.

2. (Like it was addressed in Fig 3) many of the changes in vitro assays in Fig 1 & 2 may be due to changes in T cell populations (not due to T cell intrinsic changes). Therefore, it would be important to check the T cell heterogeneity pre vs post treatments (e.g. mitochondria phenotypes, GLUT expression, integrins & chemokine expression, etc).

Following the Reviewer's suggestion, we quantified these parameters under multiple conditions. Overall, we found that:

Fig. R1. Expression of the indicated proteins on CD8+ T cells cultured for 1h (acute) in complete medium or medium without glucose or glutamine (n=4).

-) The acute deprivation of glucose or glutamine (whose effects on motility was assessed in Fig. 1B-C) had no impact on mitochondrial mass, mitochondrial membrane potential (MMP), mtROS amount and GLUT1 expression (see the **new Fig. S1C** and lines 93-95) nor on the expression of a variety of cell adhesion molecules and chemokine receptors in CD8+ T cells (see **Fig R1**).

Fig. R2. Expression of the indicated proteins on CD8+ T cells cultured in presence of the indicated drugs (n=4).

-) The addition of several metabolic drugs (whose effects on motility was assessed in Figs. 1E-G-H, i.e., 2DG, 6,8bis-Octanoic Acid, etomoxir and CB-839) had no impact on mitochondrial mass, mitochondrial membrane potential (MMP), mtROS amount and GLUT1 expression (see the **new Fig. S1H** and lines 122-125) nor on the expression of a variety of cell adhesion molecules and chemokine receptors in CD8+ T cells (see **Fig. R2**). The only exception was an increase in mtROS amount upon 6,8bis-Octanoic Acid treatment which has previously been described as one of the mechanisms by which this drug inhibits the OGDH enzyme (PMID: 24612826).

3. The authors, while testing the effects of many key metabolic inhibitors and limited glucose and glutamine, never examined the effects of hypoxia vs normoxia. The experiments strongly implicate a reliance on the TCA cycle and ATP production driven by glycolysis and glutaminolysis, which requires oxygen. All may be limited in most TME's. Although many in vitro conditions are slightly hypoxic, the authors could determine the effects of rapamycin treatment on motility in more controlled conditions of hypoxia, or perhaps by the hypoxia-mimetic CoCL2. In most experiments, T cells were either overlaid onto tumor slices or recovered shortly after injection within tumors and then imaged presumably in normoxia. In this method, it is not possible to determine how microenvironmental factors, such as hypoxia, are influencing motility.

We agree with the Reviewer that hypoxia is an important parameter to consider in our experiments and that our ex vivo conditions using tumor slices may not allow us to efficiently assess the impact of hypoxia on T cell migration. We believe that CoCL2 treatment would not be appropriate for this purpose as it is a pharmacological approach to activate the hypoxia-inducible factor HIF-1a but does not generate a hypoxic state in cells. To circumvent this problem, we generated a home-made system to test the motility of control or rapamycin-treated CD8+ T cells in collagen gel under hypoxia. In brief (and also described in more details in lines 696-701), we used a N₂ gas bottle to deliver 100% N₂ into the motility medium, which was then transferred in contact with the collagen gel using a perfusion system integrated into our spinning-disk microscope. Using an oximeter, we were able to measure the O₂ concentration, which was around 5% close to the collagen gel (i.e., mild hypoxia). Under these conditions, we observed that (i) hypoxia significantly reduces CD8+ T cell 3D motility and (ii) rapamycin treatment was ineffective in increasing cell motility under limiting oxygen concentration (see the **new Fig. S6E** and line 335). These data are consistent with what we observed in the presence of the ATP synthase inhibitor oligomycin (see old Fig. 5E). As detailed in the discussion (see lines 575-584), we plan to develop and to test novel approaches enabling T cells to move actively also in hypoxic tumor regions.

4. Additionally, the authors do not really address the T cell intrinsic factors such as smGTPases, MTOC and cytoskeletal rearrangements in the context of these treatments which would strengthen the (direct) connection between metabolism and migration compared to motility.

Following the Reviewer's suggestion, we performed the following experiments:

-) first, we analyzed the level of polymerized actin (phalloidin staining) and tubulin (antibody staining in PHEM buffer), as well as the levels of phosphorylated myosin light chain 2 (pMLC2) in CD8⁺ T cells migrating in collagen gel in the presence or absence of oligomycin (at high doses inhibiting both ATP and mtROS production). As shown in the new Fig. S4I (and lines 300-309), we observed that oligomycin treatment significantly reduced the amount of polymerized actin and tubulin as well as pMLC2 levels in migrating cells. On the contrary, we observed no differences in MTOC levels, suggesting that the assembly of this structure was unaffected by oligomycin treatment in collagen gel. Overall, these data suggest that mitochondrial metabolism has a direct impact on the polymerization of cytoskeleton components as well as the actomyosin contractility required to maintain T cell motility in a 3D environment.

-) secondly, we investigated whether rapamycin treatment, which increases mitochondrial metabolism in CD8⁺ T cells, also had an impact on actomyosin contractility. Remarkably, we observed that rapamycin treatment increased the level of pMLC2 in CD8⁺ T cells migrating in collagen gel (see the new Fig. S6F and lines 336-338), while not affecting the total levels of several small GTPases (see Fig. R3), suggesting enhanced actomyosin contractility to support active migration.

Fig. R3. Expression of the indicated proteins in ctrl- and Rapa-CD8 T cells (representative images from 1 blood donor, n=2/4 blood donors)

5. The authors show that rapamycin treatment of CAR T cells may increase CAR T cell infiltration into tumor islets in either s.c. or lung metastasis models. The authors however do not clearly show that the enhanced intratumoral motility is due to the increased mitochondrial metabolism of these cells. Is this infiltration directly due to the changes in metabolism, or is it something else, for instance rapamycin treatment changed the differentiation status of the T cell product. It may also influence the chemokine receptor profile. The authors should compare the expression of key homing markers on their control and rapamycin treated CAR T cells. Perhaps limiting mitochondria during expansion would have the opposite effect.

We agree with the Reviewer that altered expression of adhesion molecules and/or chemokine receptors upon rapamycin treatment may contribute to explain the enhanced infiltration of these cells independently to the rapamycin's metabolic effects. To test this hypothesis, we assessed the levels of several cell adhesion molecules (CD62L, CD49d, CD2, CD44, CD103, CD11a, LPAM-1) and chemokine receptors (CCR5, CXCR3, CXCR4, CCR7, CCR3) in ctrl-CD8^{CAR} and Rapa-CD8^{CAR} cells. As shown in the new Fig. S10G (see also lines 478-483), we observed no differences in the expression of these proteins between the two populations. We believe this may reinforce the importance of the rapamycin-induced metabolic reprogramming in promoting enhanced infiltration of CD8⁺ CAR T cells in our two preclinical tumor models.

6. A direct comparison of ctrl vs rapamycin treated cells via competition assays *in vivo* would strongly support the findings in this paper.

We thank the Reviewer for her/his suggestion. Following her/his recommendation, we performed a competition assay by injecting eFluor670 (eF670)-labelled ctrl-CD8^{CAR} cells and CFSE-labelled Rapa-CD8^{CAR} cells mixed in a 1:1 ratio into NSG mice bearing either lung or s.c. tumors. As shown in the new Figs. S10B and S10D, we confirmed that Rapa-CD8^{CAR} cells accumulated significantly more efficiently into tumor islets in the lung tumor model (see the new Fig. S10B and lines 441-444) and infiltrated better into the s.c. tumor mass (see the new Fig. S10D and lines 463-466) compared to ctrl-CD8^{CAR} cells.

7. There is a difference in tumor islet infiltration compared to solid tumor infiltration and the language is not always clear. Would have been nice to see characterization of hypoxia and nutrient density in tumor islets.

We apologize to the Reviewer for our lack of clarity. We have revised the text to correct the language in the section describing *in vivo* data (related to Figure 7).

We agree that hypoxia may represent an obstacle to proper infiltration of CAR T cells into solid tumors, as already reported in the literature (PMID: 30885258). This would be in line with our new results, following the

Reviewer's suggestion, showing that the rapamycin-mediated increase in CD8⁺ T cell motility in collagen gel is lost under hypoxic conditions (see the **new Fig. S6E**). However, we believe that the two *in vivo* models used in our study are not suitable for studying the impact of hypoxia on CAR T cell infiltration. In the lung tumor model, the oxygen tension in the mouse lung is most likely too high for the generation of hypoxic tumor areas, also taking into account the fact that tumor islets are relatively small (see images in Fig. 7B). To verify the presence of hypoxia in our subcutaneous tumor model (A549 cells injected s.c. into NSG mice), we stained tumor slices with LDHA, one of the metabolic markers best correlated with hypoxia in tumors (PMID: 21569415). However, we found that LDHA expression was significantly lower than that observed in tumors derived from s.c. injection of human BxPC3 tumoral cells into NSG mice (see **Fig. R4**). We believe that the most plausible explanation is that BxPC3 tumors grow significantly bigger than A549 tumors, which remain too small to develop clear hypoxic niches. Therefore, we believe that our A549 s.c. tumor model is probably not the best one to study the role of hypoxia in T cell infiltration. Given the importance of this topic, as detailed in the discussion (see lines 575-584), in the near future we would like to better characterize the impact of hypoxia on T cell infiltration in order to develop new approaches to sustain infiltration in hypoxic tumor regions. However, it will be preferable to move on to more hypoxic tumor models, such as BxPC3-derived tumors.

Fig. R4. Representative immunofluorescence images of LDHA and EpCAM expression in tumor slices derived from s.c. injection of BxPC3 or A549 cells into NSG mice.

Last, we agree with the Reviewer that a characterization of nutrient density in tumor islets in correlation with T cell motility would be extremely interesting. However, we admit that we are currently lacking the tools to perform such a combined analysis. Indeed, although some approaches are emerging to characterize nutrients in tumor areas, they require fast digestion and/or dissociation of tissue to rapidly isolate nutrients and are therefore incompatible with the approaches currently used in our laboratory to assess T cell intra-tumoral motility (requiring several steps for preparation, staining and imaging of tumor slices and therefore altering nutrient distribution within the tumor slice).

Reviewer #2 (Remarks to the Author):

The manuscript submitted by Simula et al., sheds new light into our understanding of metabolic reprogramming and T cell motility. This work will appeal to the adoptive immunotherapy audience as the ability of T cells to infiltrate solid tumors and reach cancer cells is a critical aspect of efficacy. In an important advance, the authors uncover the relative importance of glycolysis versus OX PHOS in proliferation versus motility, respectively. Distinct metabolic requirements of T cell responses were previously unclear as it was assumed that overall metabolic flexibility would support all needs. The authors include excellent models such as 3D collagen gels and tumor slices to demonstrate the preferential reliance on mitochondrial metabolism on motility within tumors. Remarkably, the paper reveals a novel role of lactate in T cell motility (in medium lacking glutamine and conditioned with 2DG). These observations should be extended to pyruvate, and acetate. As they can support mitochondrial metabolism when glucose is low, can they recover motility in 2DG-treated cells cultured in glutamine-free medium?

We thank the Reviewer for her/his general appreciation of our work. We apologize for not having clearly highlighted in the first version of our manuscript that we also tested the ability of acetate and pyruvate (as well as DM-succinate and DM-alpha-ketoglutarate) to restore motility in 2DG- and oligomycin-treated CD8⁺ T cells cultured in glutamine-free medium. Similar to lactate, we found that all these nutrients were able to rescue 2DG- but not oligomycin-mediated inhibition of cell motility (see **Figs. S2E-F**), further reinforcing the role of OXPHOS in supporting CD8⁺ T cell motility.

Alternatively, it would be interesting to see if MCT1 inhibition reverses the beneficial impact of lactate.

Following the Reviewer's suggestion, we tested whether MCT1 inhibition could reverse the beneficial effect of lactate. As shown in the new Fig.S2D (see also lines 177-181), we found that lactate rescued the 2DG-mediated motility inhibition of cells cultured in glutamine-free medium, but its effect was lost in the presence of the MCT1 inhibitor AZD3965. This result confirms that lactate is actively taken up by T cells (and presumably used in OXPHOS) to sustain cell motility upon inhibition of glycolysis.

In another advance, the authors develop a model in which the contingency energy (respiratory) reserve supports T cell motility in tumor slices. By impairing maximal respiratory function with OA, the migratory capacity is limited providing a novel link between the contingency energy reserve in the mitochondria and T cell migration. IL-7/15 which are increasingly used to condition T cells for cell-based therapies, potentiate mitochondrial function and this carries over to enhanced motility as expected. Here the authors show that T cells that can effectively infiltrate tumor islets demonstrate enhanced expression of IDH2, ATP5a and TOM20 exemplifying the importance of mitochondrial metabolism in T cells within tumors. Overall the paper has the potential to be translationally relevant, and scientifically appealing but there are a number of weaknesses that need to be addressed.

Main Points

Given the mechanism of action of CB839, and its selectivity for GLS-1, the data shows that T cells with an intact GLS-2 isoform cannot compensate sufficiently to support motility. Can the authors discuss this in more detail? And can they show what happens to max oxygen consumption rates/mitochondrial function when CB839 is employed?

We would like to thank the Reviewer for this question, which has allowed us to better clarify the role of GLS1 and GLS2 in supporting T cell motility. Indeed, GLS2 is much less expressed in activated CD8⁺ T cells than GLS1, (see Schmiedel dataset at <https://www.proteinatlas.org/ENSG00000115419-GLS/immune+cell#top> for GLS1 and <https://www.proteinatlas.org/ENSG00000135423-GLS2/immune+cell> for GLS2). However, to further clarify the role of these two enzymes, we compared the effects of the GLS1 inhibitor CB-839 and the GLS2 inhibitor Compound-968 in regulating activated CD8⁺ T cell mitochondrial metabolism (oroboros measurement) and motility in collagen gel. In agreement with the expression data, we found that only inhibition of GLS1, but not GLS2, significantly reduced mitochondrial respiration (particularly maximal and spare respiratory capacities, see new Fig. S1E and lines 107-113). These data suggest that either GLS2 does not support mitochondrial respiration (since its expression level is too low) or that GLS1 can compensate for it. On the contrary, GLS1 is required to sustain mitochondrial respiration, as GLS2 does not compensate for its activity. In line with these metabolic data, we found that only inhibition of GLS1, but not GLS2, significantly reduced CD8⁺ T cell migration in collagen gel (see new Fig. S1F and lines 107-113). Overall, these data suggest that GLS1 plays a prominent role in supporting T cell motility.

In figure 7D, the anti-tumor function of rapa- CAR T cells are analyzed in an in vivo xenograft model (A549 IV infused). There are inherent limitations with this model as the standard CAR performs so poorly and the differences in tumor control of rapa-treated CAR T cells are not obvious. The numbers of untreated mice (n=2), CAR (n=3) and rapa CAR (n=3) are very low for an in vivo experiment. It's also unclear if this model was repeated more than once? The tumor model needs to be repeated.

Following the Reviewer's suggestion, we repeated the bioluminescence experiment (A549 i.v. lung tumor models + CAR T cells) to increase the number of mice taken into account. The updated results (n=6 untreated mice, n=8 mice treated with ctrl-CD8⁺ CAR T cells, n=8 mice treated with Rapa-CD8⁺ CAR T cells) are now shown in the new Fig. 7D). Moreover, in order to highlight the potential beneficial effect of rapamycin treatment, we have injected a suboptimal amount of CAR T cells (5x10⁵ cells, see line 448 and Materials & Methods line 912) which are not able to control tumor growth in the ctrl-CD8^{CAR} condition.

In supplementary figure 10 (relating to Figure 7), looking at tumor growth in individual mice, 4/6 of the untreated mice have similar levels of tumor growth as their CAR counterparts. It's also unclear if this experiment was repeated more than once. A simple killing assay in vitro would be informative to determine if rapa-treated EGFR CAR T cells have enhanced tumor control. Additionally, a measurement of EGFR expression in A549 cells (target cell line) would be beneficial as overexpression maybe needed to boost CAR-mediated cytotoxicity.

The experiment in Fig. 7F (monitoring tumor growth in the "A549 s.c. tumors + anti-EGFR CAR T cells" setting) was repeated twice and the total number of mice used is indicated in the figure legend. We believe that the high variability observed in the growth curves of mice (see

Fig. R5. Mean growth curves of A549-derived s.c. tumors in NSG mice untreated (black) or inoculated with ctrl- (red) or Rapa-CD8^{CAR} T cells (green) in three different mouse cages (data included in the experiment reported in Figure 7F).

Fig. S10E for individual growth curves) is mainly due to a cage effect. Indeed, for each cage, we found that ctrl-CD8^{CAR} T cells poorly controlled tumor growth compared to untreated mice, while tumor growth of "Rapa-CD8^{CAR} T cell" mice was always lower than that of untreated controls. However, we observed great variability in tumor growth in untreated mice from cage to cage, which was also reflected in mice treated with ctrl- or Rapa-CD8^{CAR} T cells (raw measurements from three different cages in Fig. R5 as an example). We apologize to the Reviewer for not clearly highlighting in the first version of the manuscript that we have added an additional panel to account for this cage-effect difference. Indeed, in the Fig. S10F, we have reported the relative growth of ctrl- or Rapa-CD8^{CAR} T cell-treated mice normalized to the growth of untreated mice for each cage (*i.e.*, growth of untreated mice is set as "1" for all time points and for each cage), in order to better highlight the differences observed in treated mice and to remove the large variability observed between tumor growth rate in untreated mice from different cages.

Fig. R6. A549 cells have been left unstained (red) or stained with anti-EGFR antibody (blue).

We also apologize to the Reviewer for not clearly pointing out in the first version of our manuscript that an *in vitro* killing assay (against EGFR⁺ A549 cells) was indeed performed (see old Fig. 6I). The results indicate that Rapa-CD8^{CAR} T cells exhibit a cytotoxicity similar to that of ctrl-CD8^{CAR} T cells with only a slight reduction in the killing rate at low effector:target ratio. In addition, our data confirm the good expression of EGFR at the surface of the A549 cells (see Fig. R6). Moreover, additional *in vitro* killing experiments, not included in the manuscript, (see Fig. R7) confirm that while normal T cells (*i.e.*, not expressing the CAR molecule) do not recognize and kill A549 tumor cells, CAR-expressing T cells are very efficient at killing A549 cells in a dose dependent manner.

Fig. R7. *In vitro* killing assay of A549 cells cultured alone (yellow) or co-cultured in presence of normal (*i.e.*, not CAR) T cells (blue) or anti-EGFR CAR T cells at different doses (pink to red). Measurements of A549 viability were obtained through incucyte.

It would be interesting to see if OA leads to a reduction in TCA cycle metabolites acetyl CoA, citrate, succinyl CoA.

Following Reviewer's suggestion, we have measured the levels of acetyl-CoA, citrate, and succinate in

activated CD8⁺ T cells treated or not with 6,8bOA for 1h. As shown in the **Fig. R8**, we found that 1h treatment of 6,8bOA had no impact on the levels of acetyl-CoA and citrate (although the latter showed a trend towards an increase), while it significantly reduced succinate levels. We believe that these results are consistent with what we observed upon PDHA1 and OGDH knock-out through Crispr/Cas9 approach (see Fig. 1J and Fig. S1M). Indeed, PDHA1 KO had no impact on mitochondrial respiration (neither basal nor maximal capacity) (see Fig. 1J and Fig. S1M), suggesting that PDHA inhibition (also through 6,8bOA) can be compensated to maintain TCA cycle activity and therefore acetyl-CoA levels (maybe sustained through an enhancement of fatty acid oxidation) as well as downstream citrate. On the contrary, OGDH KO significantly reduced both basal and maximal respiration (see Fig. 1J and Fig. S1M). This is suggestive of a TCA cycle blockade at OGDH level upon OGDH inhibition. Thus, downstream succinate levels should be reduced (as we observe with 6,8bOA). We hypothesize that this may also explain the tendency of upstream citrate to slightly accumulate in cells).

Fig. R8. (6,8bOA / ctrl) ratio of the levels of indicated metabolites measured in activated CD8⁺ T cells treated or not with 6,8bOA for 1h (n=4).

The conditioning strategy at 39°C is intriguing. How does activation at 39°C impact subsequent CAR transduction? i.e. it more/less permissive for CAR lentiviral infection?

Following the Reviewer's suggestion, we measured the CAR transduction rate in cells cultured at 37°C vs 39°C. As reported in the new **Fig. S11A** (see also lines 495-496), T cell activation at 39°C did not impact on the efficiency of CAR expression.

Minor Points

As described above, it's unclear from the figure legend how often the xenograft models were performed.

Both the "A549 i.v. + CAR T cells" (Fig. 7D) and the "A549 s.c. + CAR T cells" experiments (Fig. 7F) have been repeated twice. The total number of mice used for each experimental condition is reported in the corresponding figure legends.

The authors refer to 6,8 bis(benzylthio) octanoic acid as octanoic acid in the materials section. I think this will cause confusion in the main text. The former is an inhibitor of PDH and OGDH; the latter is an 8-carbon medium chain FA species which fuels the TCA cycle.

We apologize to the Reviewer for our excessive attempt to simplify acronyms which generated confusion. We have now redefined our inhibitor with the acronym 6,8bOA in the text and figures.

An important point for future studies relates to the toxicity, and variability of CCCP for human T cell cultures. A shift to BAM15 as an uncoupling agent would be preferable.

We understand the Reviewer's concern, as BAM-15 is currently preferred over other uncoupling agents for metabolic studies with T cells. To exclude potential side effects of CCCP in our Oroboros measurement, we repeated the assessment of mitochondrial respiration between control (ctrl-CD8) and rapamycin-treated (Rapa-CD8) CD8⁺ T cells (probably the most important metabolic difference we report in the paper to support our findings) by using BAM-15 instead of CCCP. As the reviewer can appreciate in **Fig. R9**, we found that using BAM-15 gave similar results to those observed with CCCP (compare **Fig. R9** with Fig. 5C in the manuscript).

Fig. R9. Oroboros measurements for basal, maximal, and spare respiratory capacity in ctrl-CD8 and Rapa-CD8 T cells using BAM-15 instead of CCCP as uncoupling agent (n=4).

Supplementary figure 1 (A) it looks like the largest fold increase occurred without glucose and glutamine. Is this mislabeled?

We apologize for this error. The left condition is indeed complete medium (with both glucose and glutamine). We have corrected it in the figure. Thanks to have caught this mistake.

In supplementary figure 10 (looking at the individual growth curves), 4/6 of the untreated mice have similar levels of tumor growth as their CAR counterparts.

Please see our answer to the same question above.

In supplementary figure 10C, the legend refers to the xenograft model in Figure 7E. It's actually referring to Fig. 7F.

We are sorry for our mistake. We have corrected it in the figure. Thank you for pointing it out.

I think it is important to note that the scenario described in this paper (such as the mitochondrial adaptations observed in IL-7/15 treated T cells) and potentiated mitochondrial function, maybe context specific (lung cancers). Oxygen levels in the lung may permit/support mitochondrial function that in turn facilitates enhanced motility. In other models such as melanoma, and ovarian cancer, mitochondria function is severely suppressed, particularly in the CD8 T cell subset. As noted in their discussion, this paper will seed future efforts to address how motility can be supported in hypoxic TMEs such as these. Mindful to add, glucose uptake is also severely inhibited in TILs (B16 melanoma models). From this perspective, exploring how other fuels such as pyruvate, acetate, and lactate support motility are important future directions.

We thank the Reviewer for her/his comment. We fully agree that our models are not suitable for studying the effects of hypoxia on T cell infiltration, as indeed we also indicated in response to a comment from the Reviewer 1 (please see **Fig. R4**). We have measured *in vitro* motility of T cells under mild hypoxia (around 5% O₂ tension) and we found that under this condition rapamycin-treatment was unable to increase cell motility, which is consistent with what has been reported with oligomycin (see old Fig. S5E and the new **Fig. S6E**). As mentioned in the discussion, we now plan to study how hypoxia could be counteracted to sustain T cell infiltration into hypoxic tumor areas (see lines 575-584). However, we will have to switch to different preclinical models (see also our response to Reviewer 1 for more details). Last, we thank the Reviewer for her/his suggestion about the potential clinical application of nutrients bypassing glycolysis to support OXPHOS. We have added a sentence in the Discussion to include this aspect (see lines 569-572).

Reviewer #3 (Remarks to the Author):

This manuscript investigates the metabolic requirements for T cell migration in solid tumors. Overall, there is a massive amount of experiments and data in this paper. While the differences are not dramatic, many are significant, and multiple inhibitory approaches are taken, to support a link between mitochondrial metabolism and T cell migration in solid tumors. Ultimately, much of the data supports that enhancing ATP and mitochondria-based metabolism supports T cell migration in human tumor slices and xenograft tumor models. Two previously reported approaches (rapamycin or heat treatment of engineered T cells prior to adoptive cell therapy) known to promote mitochondrial fitness are test here and are shown to enhance T cell migratory capacity in tumor nests and ultimately antitumor activity. Overall, the data is supportive of the hypothesis and the authors are fair in their interpretations. Some issues remain including model and system choice as well as novelty overall.

We would like to thank the Reviewer for her/his general appreciation of our work.

Major Issues

Many of the results provided are correlative and/or supportive but not clearly mechanistic in nature.

We agree with the Reviewer that our study presents limited mechanistic insights. Our work is mainly dedicated to (i) understanding how different metabolic pathways are exploited by T cells during their intra-tumoral migration (Figs. 1-3) and (ii) developing novel strategies to improve T cell motility in human solid tumors (Figs. 5-8). However, we also highlighted in Figure 4 that mitochondrial metabolism supports CD8+ T cell 3D motility not only through ATP production but also via a modulation of mtROS production (Fig. 4). We believe that this is an important point, as the role of ROS in regulating 3D amoeboid motility is still debated (PMID: 20822528 and 24486154). Here, our data reveal that mtROS are required to sustain motility, but if their accumulation concomitantly impairs ATP production this would drive a reduction in cell motility (see Fig. 4). Furthermore, following the Reviewer's suggestion and the advice from Reviewer 1, we investigated whether metabolic manipulations could have an impact on the cytoskeleton of migrating CD8+ T cells. Our data indicate that oligomycin treatment (which impacts both ATP levels and mtROS production at high doses, see Fig. 4B-C) significantly reduces the amount of polymerized actin (phalloidin staining) and tubulin (antibody staining in PHEM buffer) in CD8+ T cells migrating within a collagen gel (**new Fig. S4I** and lines 300-309). A reduction in the level of phosphorylated myosin light chain 2 (pMLC2) was also observed (**new Fig. S4I** and lines 300-309). These data suggest that mitochondrial metabolism has a direct impact on the ability of migrating cells to correctly rearrange their cytoskeleton and energetically support their actomyosin contractility. These results provide a further mechanistic insight into the metabolic regulation of CD8+ T cell 3D motility.

No immunocompetent animal models are tested and thus the restricted use of xenografts is a limitation.

We agree with the Reviewer that our NSG mouse model, although regularly used as a preclinical model to test for CAR T cell efficacy, does not allow us to investigate whether rapamycin treatment can promote CAR T cell infiltration in an immunocompetent animal model. To address Reviewer's request, we performed a preliminary experiment to test whether rapamycin treatment is effective in promoting infiltration of murine effector OT-I CD8 T cell into a s.c. tumor model in immunocompetent C57BL/6 mice. First, we confirmed that rapamycin treatment promotes migration of OT-I T cells in a collagen gel (see **Fig. R10**). Next, we injected control or rapamycin-treated OT-I cells into immunocompetent mice bearing B16-OVA tumors. After 2 days, we assessed the level of infiltration of OT-I cells into the tumor mass by flow cytometry. Our results indicate that rapamycin-treated OT-I cells infiltrated the tumor mass significantly better than control OT-I cells (see **Fig.**

Fig. R10. Motility in collagen gel of CD8+ OT-I cells cultured or not in presence of rapamycin (n=3).

Fig. R11. Infiltration of murine ctrl or Rapamycin-treated OT-I CD8+ T cells into the tumor mass derived from inoculation of B16-OVA cells into immunocompetent c57 mice (n=5 ctrl; n=10 rapa).

R11). These results suggest that rapamycin treatment may also be effective in promoting CD8+ T cell infiltration in an immunocompetent animal model. Although encouraging, we prefer not to include these data in the manuscript but to provide them for the Reviewer's appreciation in this response letter. Indeed, since the current manuscript contained a huge amount of data on human T cells, we would like to better characterize the effects of rapamycin

treatment in the immunocompetent OT-I murine model in a future dedicated study.

Text requires major editing for conciseness and correct grammar, tenses, etc.

We apologize to the Reviewer for our grammatical errors and lack of conciseness. The text has now been revised to improve clarity and soundness.

The authors likely test every inhibitor available to target various metabolic pathways in T cells. Is there a way the authors can make the text and figures more linear and understandable to a broader audience. Is it

necessary to show all the negative data in the main figures and restrict labels to acronyms of inhibitors? Or can the data be organized in a more logical manner with headings for specific pathways targeted.

Following the Reviewer's suggestion, we have revised our main figures to improve the clarity of the presentation, in particular to better identify the targeted pathways (see for examples the new headings in Fig. 1E-G and Fig. S1E-F, new diagrams in Fig. 1I, 2A, 2C, 2G, 4F, 4J, and 4N). We believe that "negative" data regarding the role of fatty acid oxidation in supporting T cell motility deserve a place in the main figures, as the concept that FAO is dispensable for migration is an important point in our study (*i.e.*, mitochondrial metabolism promoted by glucose and glutamine but not FAO supports intratumoral motility of CD8⁺ T cells).

Assessment of migration of polyclonal, non-tumor reactive T cells in a tumor microenvironment is a limitation in the early part of this study and is somewhat a limitation. Unclear why non-specific T cells are being evaluated.

We believe that assessing the motility of non-tumor reactive T cells in the early part of the study helps us better define the intrinsic role of metabolism in regulating T cell migration without the possible confounding effect of concomitant TCR engagement (which may act as a stop signal for motile T cells).

Are the metabolic properties identified general principles for effector T cell migration regardless of the tumor microenvironment?

We thank the Reviewer for this relevant question. Our laboratory is expert in the study of intra-tumoral T cell motility and our objective is to characterize the metabolic regulation of T cell migration and to develop metabolic strategies to reinforce it. Therefore, our experiments (and our interpretations of the data obtained) were limited to the tumor context. That said, we agree that our results may have a broader impact that is not necessarily limited to the tumor microenvironment. Indeed, our experiments in collagen gels (2mg/ml) suggest that mitochondrial metabolism can enhance T cell motility in *any* dense environment (not necessarily a tumor environment). To reinforce this point, we measured the motility of control and rapamycin-treated CD8⁺ T cells into collagen gels of different densities. Remarkably, we observed that the rapamycin-mediated enhancement of mitochondrial metabolism effectively increases cell motility only in *denser* environments, whereas at lower densities T cells have sufficient energy to move rapidly (see Fig. R12). In line with this, we found that CAR T cell transmigration across an endothelial layer (mimicking blood extravasation) is significantly increased by rapamycin treatment (see old Fig. 6E). This may suggest that T cell recirculation within lymphoid organs can be increased by rapamycin, as we have indeed observed (see Fig. R13). Overall, we believe that these data may be potentially important in a context of a pathogenic infection (viral, bacterial) or even auto-immunity. However, we feel that these topics are beyond the scope of the current manuscript, and we don't have the expertise, tools, as well as ethical permissions to perform experiments using auto-immune or viral models. However, we would be happy to develop these new lines of research in the future by establishing new collaborations with experts in these fields.

Fig. R12 Motility of control or rapamycin-treated human CD8⁺ T cells into collagen gel at the indicated densities (n=12).

Fig. R13 Relative ratio of human rapamycin-treated vs control T cell infiltration into spleen 24h after injection at 1:1 ratio into NSG mice (n=5).

Is it not already known that cellular migration requires ATP? This perhaps is not surprising.

We agree with the Reviewer that the concept of cell migration requiring ATP is not *per se* novel nor surprising. However, we believe that our study provides two additional important insights that help to better understand how metabolism controls cellular migration:

-) firstly, we investigated how different nutrients are used by T cells to generate the energy (*i.e.*, ATP) required for their motility (*i.e.*, nutrient preferences). Indeed, our findings established that mitochondrial-derived ATP and not glycolytic one plays a major role in supporting T cell motility. Moreover, mitochondrial

ATP is mainly generated by glucose and glutamine oxidation but not by fatty acids. Overall, we believe that these findings are important (*i*) to better understand how metabolic manipulations (quite popular now as anticancer approaches) can impact on intra-tumoral T cell motility and (*ii*) to better define metabolic strategies to enhance T cell infiltration/migration into solid tumors.

-) secondly, our data also suggest an ancillary role for mtROS in supporting motility in addition to mitochondria-produced ATP, further defining why mitochondria are important for T cell motility.

The approaches to enhance mitochondria metabolism in T cells are not particularly novel.

We agree with the Reviewer that our rapamycin-based strategy for improving mitochondrial metabolism is not *per se* particularly novel, since the notion that rapamycin treatment enhances mitochondrial respiration in T cells is well established. However, our goal was not to develop a new method for enhancing mitochondrial respiration. Rather, we wanted to investigate how metabolism regulates intratumoral T cell migration and demonstrate that a metabolic approach could be used to enhance T cell infiltration and motility into solid tumors. Indeed, we believe that this is a current gap in our knowledge because (*i*) many metabolic strategies are currently tested as cancer therapies but their potential impact on intra-tumoral T cell motility has never been investigated and (*ii*) although a number of metabolic approaches to reinforce T cell-based immunotherapy have been proposed in recent years, they have always been evaluated in terms of their impact on T cell cytotoxicity, differentiation status and/or long term persistence while their impact on cell migration has been mostly ignored.

In summary, we believe that our study advances the field forward by providing two major notions. First, by showing how mitochondrial metabolism supports the ability of T cells to migrate and infiltrate tumor islets, we provide an additional layer to explain the efficacy of antitumor strategies targeting mitochondria in T cells beyond their well-established effects on T cell persistence (which is the only explanation currently proposed in the field). Second, from a clinical perspective, we have developed a simple *in vitro* pharmacological approach generating CAR T cells with superior ability to infiltrate human solid tumors that has a strong translational potential, as it could be easily implemented in current CAR T cell manufacturing protocols. To our knowledge, this is the first immunometabolic approach against solid cancers focused on improving the intratumoral motility of CAR T cells.

While much of the results are significant, many graphs are either not significant or barely significant.

In this revised version, we have provided all the source data files (in excel format) showing the data used to generate all the graphs in the manuscript. For each figure, the data are accompanied by the statistical test performed (with exact p-values indicated).

Authors may consider to temper the interpretations and replace words like “indicate” with “support”

We thank the Reviewer for this suggestion. We have revised the text accordingly throughout the manuscript (see for example lines 153, 191, 239, 381, 513).

REVIEWERS' COMMENTS

Reviewer #1 (Remarks to the Author):

I am satisfied with the reply of the authors. In principle, all the necessary changes and experiments have been made or performed, respectively.

Reviewer #3 (Remarks to the Author):

The authors were very responsive to the criticisms and the paper is improved overall. This is a nice study.

Reviewer #1 (Remarks to the Author):

I am satisfied with the reply of the authors. In principle, all the necessary changes and experiments have been made or performed, respectively.

We thank the Reviewer for his/her suggestions that significantly improved the revised version of the manuscript.

Reviewer #3 (Remarks to the Author):

The authors were very responsive to the criticisms and the paper is improved overall. This is a nice study.

We thank the Reviewer for his/her suggestions that significantly improved the revised version of the manuscript.